# Elucidation of a four-site allosteric network in fibroblast growth factor receptor tyrosine kinases

Huaibin Chen[1†], William M Marsiglia[2†], Min-Kyu Cho[2], Zhifeng Huang[3], Jingjing Deng[4], Steven P Blais[4], Weiming Gai[1], Shibani Bhattacharya[5], Thomas A Neubert[4], Nathaniel J Traaseth[2*], Moosa Mohammadi[1*]

[1]Department of Biochemistry and Molecular Pharmacology, New York University School of Medicine, New York, United States; [2]Department of Chemistry, New York University, New York, United States; [3]Wenzhou Medical University, Wenzhou, China; [4]Skirball Institute of Biomolecular Medicine, New York University School of Medicine, New York, United States; [5]New York Structural Biology Center, New York, United States

**Abstract** Receptor tyrosine kinase (RTK) signaling is tightly regulated by protein allostery within the intracellular tyrosine kinase domains. Yet the molecular determinants of allosteric connectivity in tyrosine kinase domain are incompletely understood. By means of structural (X-ray and NMR) and functional characterization of pathogenic gain-of-function mutations affecting the FGF receptor (FGFR) tyrosine kinase domain, we elucidated a long-distance allosteric network composed of four interconnected sites termed the 'molecular brake', 'DFG latch', 'A-loop plug', and 'αC tether'. The first three sites repress the kinase from adopting an active conformation, whereas the αC tether promotes the active conformation. The skewed design of this four-site allosteric network imposes tight autoinhibition and accounts for the incomplete mimicry of the activated conformation by pathogenic mutations targeting a single site. Based on the structural similarity shared among RTKs, we propose that this allosteric model for FGFR kinases is applicable to other RTKs.

*For correspondence: traaseth@ nyu.edu (NJT); Moosa. Mohammadi@nyumc.org (MM)

†These authors contributed equally to this work

Competing interests: The authors declare that no competing interests exist.

## Introduction

Receptor tyrosine kinase (RTK) signaling fulfills fundamental functions in development, tissue homeostasis, and metabolism of metazoan organisms (*Manning et al., 2002*). The 58 human RTKs are divided into ~17 subfamilies, each featuring unique architectures in their extracellular domain that are specialized to bind and mediate the biological actions of distinct growth factors, cytokines and hormones such as EGF, FGF, PDGF, VEGF, IGF, insulin and NGF (*Lemmon and Schlessinger, 2010*). The divergent extracellular domains of RTK subfamilies are linked via a single transmembrane helix to the intracellular domain that harbors the conserved tyrosine kinase domain. Despite exhibiting major structural diversity in their ectodomains, all RTKs must rely on the universal process of ligand-induced dimerization or rearrangement of preformed dimers to elevate the intrinsic activity of the intracellular kinase domain (*Lemmon and Schlessinger, 2010*). In the case of EGFR subfamily, receptor dimerization facilitates the formation of an asymmetric kinase dimer, wherein a donor kinase allosterically stabilizes the active conformation of the receiver kinase (*Kovacs et al., 2015*). Other RTK subfamilies require trans-phosphorylation on regulatory tyrosines located in the activation loop (A-loop) in order to trigger activation. Analysis of crystal structures of unphosphorylated receptor tyrosine kinase domains representing every RTK subfamily show that in the resting state (i.e. in the absence of extracellular ligand engagement) kinase domains are autoinhibited through a variety of

**eLife digest** Many growth factors and hormones instruct cells to act in particular ways – for example, to divide, specialize or migrate – by binding to proteins called receptor tyrosine kinases on the surface of the cell. Receptor tyrosine kinases comprise a region that binds to the signaling molecules outside the cell (the receptor domain) and a region that interacts with and modifies other proteins within the cell (the kinase domain). When a growth factor or hormone binds to the receptor domain, the receptor domains of two identical receptor tyrosine kinases form a dimeric complex, bringing the kinase domains close together so that they can activate each other. This activation initiates a long chain of protein-protein interactions that leads to a cellular response.

Some mutations that occur in the kinase domain cause the kinase to frequently sample the activated state without needing to bind to a signaling molecule. This can lead to cancer or growth disorders. It is not known how these mutations allow the kinase domain to bypass the control mechanisms that prevent it from activating at the wrong time.

Fibroblast growth factor (FGF) receptor is a receptor tyrosine kinase in which many disease-causing mutations have been identified in the kinase domain. Chen, Marsiglia et al. have now used the human form of the FGF receptor to investigate how some of these mutations affect kinase structure and activity. The structures of normal and mutant forms of the FGF receptor were determined using X-ray crystallography and nuclear magnetic resonance spectroscopy. Combining these results with activity data from the diseased receptors revealed four key interdependent sites in the protein that play essential roles in maintaining the receptors in an inhibited state. Disease-causing mutations are generally located in these sites, and take advantage of their interdependence to make the kinase active more frequently, even when the receptor domain is not bound to a signaling molecule.

Chen, Marsiglia et al. suggest that these sites may act in the same way in other receptor tyrosine kinases, as these proteins have many similarities in their structures. The findings reveal important regions within the kinase domain that could be targeted by inhibitors to prevent the harmful effects of disease-causing mutations. Future studies will be needed to understand in more detail how interactions between the key sites change the activity of the kinases.

mechanisms (*Huse and Kuriyan, 2002*). In the case of insulin receptor, MUSK, and TRK subfamilies, kinase autoinhibition is mediated by the A-loop itself whereby the conserved DFG motif at the N-terminal end of the A-loop blocks ATP binding and a regulatory tyrosine at the C-terminus of the A-loop acts as a pseudo substrate competing with substrate binding (*Hubbard, 2002*). In another model based on the crystal structure of FLT3 (*Griffith et al., 2004*), the juxtamembrane region located outside the conserved kinase domain is involved in maintaining the autoinhibited kinase conformation along with the A-loop. Hydrophobic residues from the juxtamembrane insert into the ATP binding cleft to both compete with ATP binding and to facilitate the inactive conformation of the A-loop. Akin to the first model, a regulatory tyrosine in the A-loop inserts in the substrate binding site and acts as a competitive inhibitor (*Hubbard, 2002*). In the case of the FGFR, KIT, PDGFR, and VEGFR subfamilies, kinase autoinhibition is principally mediated by a network of hydrogen bonds in the vicinity of the kinase hinge region that constrains the ability of the kinase to adopt the active conformation (*Chen et al., 2007*). Additionally, in FGFR kinase an arginine at the C-terminal end of the A-loop makes a salt-bridge with the catalytic base resulting in occlusion of the enzyme's active site (*Mohammadi et al., 1996*).

Despite employing different modes of autoinhibition, all receptor tyrosine kinases adopt a common active state conformation upon A-loop tyrosine phosphorylation (*Hubbard and Till, 2000*). The available structures of A-loop phosphorylated kinases each display a salt-bridge between the phosphotyrosine and a conserved arginine residue within the A-loop that causes a dramatic structural rearrangement of the A-loop (*Huse and Kuriyan, 2002*). This conformational change sets in motion a series of poorly characterized long-range allosteric events that ultimately enables the kinase to adopt the active conformation (*Chen et al., 2007*). While crystal structures have provided atomistic insight into the two major conformational states (i.e. autoinhibited and activated forms), the role of

enzyme dynamics and allosteric communication involved in this structural change can only be indirectly inferred from the static depictions.

The FGFR subfamily of tyrosine kinases has emerged as an excellent model system to elucidate the molecular basis underlying allosteric regulation of RTKs in a physiologically relevant and unbiased fashion. There are over 30 different naturally occurring germline and somatic gain-of-function mutations that map to the kinase domain of the four human FGF receptors (FGFR1-FGFR4) with the hinge region and A-loop representing the two most common hotspots (*de Ravel et al., 2005*; *Kan et al., 2002*; *McGillivray et al., 2005*; *Zankl et al., 2004*; *Deutz-Terlouw et al., 1998*; *Grigelionienė et al., 2000*; *Mortier et al., 2000*; *Bellus et al., 2000*). These activating mutations give rise to many forms of human congenital skeletal disorders of varying clinical severity and acquired cancers (*Wilkin et al., 2001*; *Wilkie, 2005*; *Webster and Donoghue, 1997*; *Passos-Bueno et al., 1999*; *Rand et al., 2005*; *Chesi et al., 1997*; *Richelda et al., 1997*; *Cappellen et al., 1999*). Indeed, structural and biochemical analyses of gain-of-function mutations mapping to the kinase hinge region led us to identify a network of hydrogen bonds formed by three residues termed the molecular brake that suppress the ability of FGFR kinase to attain the active conformation (*Chen et al., 2007*). Mutation at each of the three constituents of the molecular brake disrupts the network of hydrogen bonds resulting in a profound increase in the intrinsic kinase activity of FGFRs and a variety of human skeletal disorders and cancers. More recently, through crystallographic and solution NMR analyses of five pathogenic FGFR2 kinases at a conserved lysine in the A-loop (the second hotspot), we showed that these mutations introduce additional hydrogen bonding or hydrophobic contacts to differentially stabilize the active conformation of the A-loop (*Chen et al., 2013*) in a dynamic two-state equilibrium regimen. In this model, FGFR kinase domain toggles between a conformationally rigid autoinhibited state and a conformationally flexible activated state. Notably, the kinase activity was found to directly correlate with the fractional population of the active state.

In all structures of A-loop mutants, the hydrogen bonds at the molecular brake, which are over 25 Å from the mutational site, were broken leading us to infer the existence of a long-range allosteric connectivity between the A-loop and molecular brake. In this report, we took advantage of an additional panel of naturally occurring pathogenic and acquired drug-resistant mutations (*Byron et al., 2013*; *Goos et al., 2015*; *Kant et al., 2015*) to chart the trajectory of this allosteric connectivity using X-ray crystallography, NMR spectroscopy and an in vitro kinase assay coupled with mass spectrometry. Our data reveal two clusters of hydrophobic interactions, termed 'DFG latch' and 'αC tether', which relay allosteric signals between the A-loop and the molecular brake. These hydrophobic networks, located strategically at the interface between the N- and C-lobes of the kinase, sense the structural fluctuations in the A-loop and molecular brake and undergo rearrangements to determine the kinase interlobe angle and hence kinase activation. Naturally occurring pathogenic and acquired drug-resistant mutations seize the four-site allosteric control pathway to bias the kinase equilibrium toward the active state thereby causing excessive signaling in human skeletal disorders and cancers. In addition to enriching our understanding of the molecular mechanisms of FGFR-related diseases, these findings are anticipated to make a major impact for ongoing drug discovery efforts targeting FGFRs and other RTKs such as PDGFRs, KIT, VEGFRs, and CSF1R that share the four-site allosteric mechanism.

## Results and discussion

### Structural analysis of pathogenic FGFR2K mutations suggests a long-range allosteric crosstalk between A-loop and the molecular brake

We recently reported the crystal structures of a panel of five pathogenic FGFR2 kinases (FGFR2K) harboring substitutions for glutamic acid, methionine, asparagine, glutamine and threonine of an FGFR-invariant lysine (K659 in FGFR2) in the A-loop, and showed that these mutations confer gain-of-function by introducing novel intramolecular hydrogen bonding or hydrophobic contacts to stabilize the active state conformation of the A-loop (*Chen et al., 2013*). Notably, in these pathogenic kinases, the hydrogen bond network at the molecular brake located 25 Å from the site of mutation was disrupted indicating that conformational changes of the A-loop propagate across the enzyme via long-range allostery (*Figure 1A*). Previously, we also solved crystal structures of pathogenic FGFR2Ks (N549H, N549T, E565A, E565G, K641R) that directly target the three constituents of the

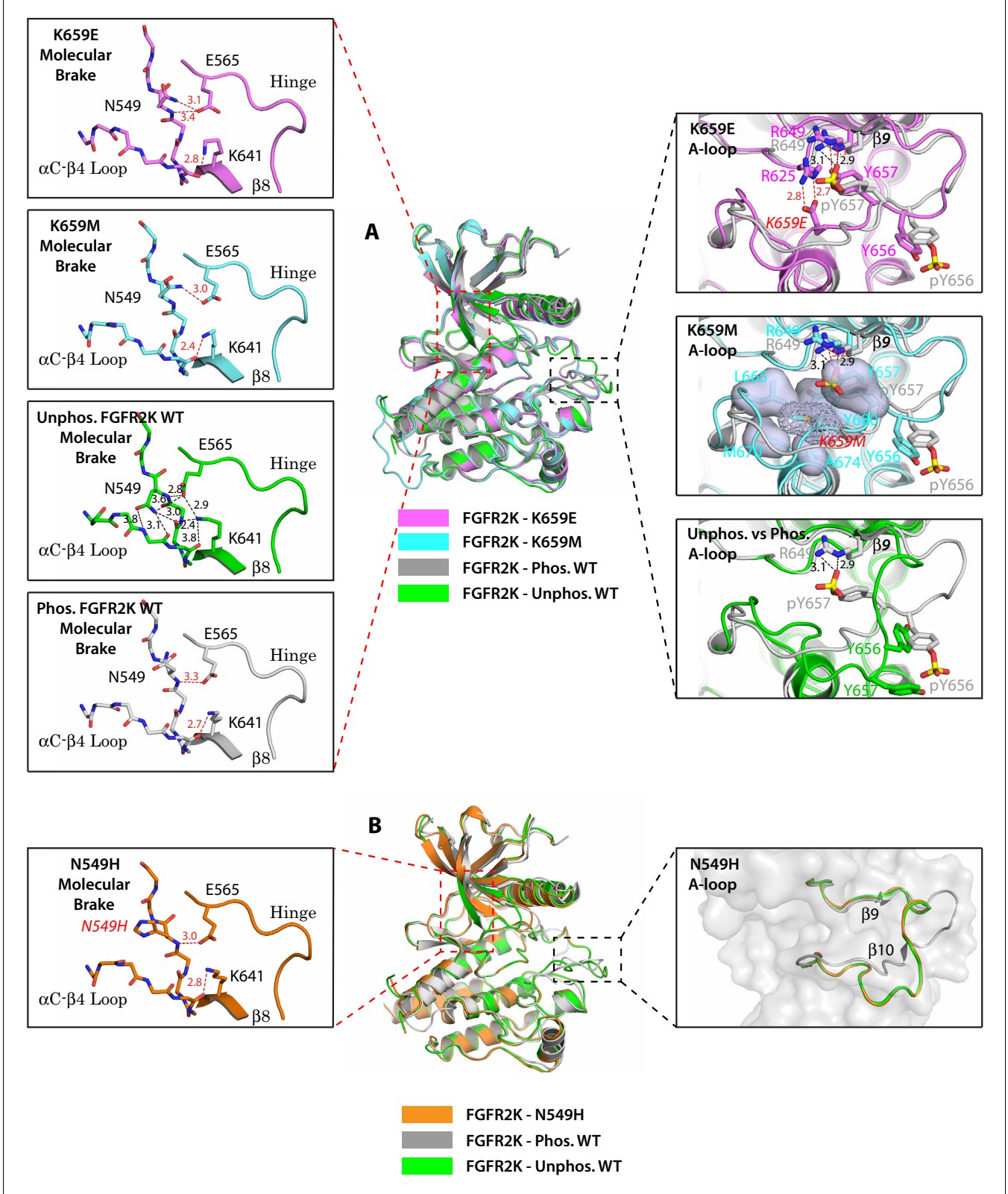

**Figure 1.** Single mutations at the A-loop cause disengagement of the molecular brake while single mutations at the molecular brake constituents are unable to drive the A-loop to the active conformation. (**A**) Overlays of crystal structures corresponding to unphosphorylated K659E FGFR2K (PDB ID: 4J97 [*Chen et al., 2013*], in magenta), unphosphorylated K659M FGFR2K (PDB ID: 4J96 [*Chen et al., 2013*], in cyan), unphosphorylated WT FGFR2K (PDB ID: 2PSQ [*Chen et al., 2007*], in green) and the A-loop phosphorylated WT FGFR2K (PDB ID: 2PVF [*Chen et al., 2007*], in grey). The insets to the

*Figure 1 continued on next page*

*Figure 1 continued*

left of the panel display close-up views of the molecular brake regions of each of the four superimposed crystal structures. The insets to the right show close-up views of the A-loops of K659E (upper inset), K659M (middle inset) and the unphosphorylated WT FGFR2K (lower inset) compared with that of the A-loop phosphorylated WT FGFR2K. The three hydrogen bonds introduced by the K659E mutation that support the active conformation of the A-loop are shown as red dashed lines. The hydrophobic interactions introduced by the K659M mutation are represented by semitransparent surfaces, and the hydrogen bonds between Y657 and R649 are shown as red dashed lines. (**B**) Crystal structure of the unphosphorylated N549H mutant of FGFR2K (PDB ID: 2PWL [*Chen et al., 2007*], in orange) superimposed onto those of the unphosphorylated WT FGFR2K (PDB ID: 2PSQ [*Chen et al., 2007*], in dark green) and the A-loop phosphorylated WT FGFR2K (PDB ID: 2PVF [*Chen et al., 2007*], in grey). The inset to the left shows a close-up view of the molecular brake region of the N549H mutant. The inset to the right shows a close-up view of the three superimposed A-loop conformations. Note that the A-loop conformation of the N549H mutant is identical to that of unphosphorylated WT FGFR2K despite the dissociation of the molecular brake by the mutation. Side chains of relevant residues are shown as sticks and the mutated residues are labeled in red letters. Relevant hydrogen bond distances (in Å) are indicated. Atom colorings are as follows: oxygens in red, nitrogens in blue, phosphorus in yellow, and carbons are colored according to the kinase structure to which they belong.

molecular brake, namely N549, E565 and K641, and showed that these mutations disrupt the autoinhibitory network of hydrogen bonds leading to profound kinase activation (*Chen et al., 2007*). Surprisingly, however, the A-loop conformation and angle between the N- and C-lobes of the kinase in these mutants remained identical to unphosphorylated wild-type (WT) FGFR2K (*Figure 1B*). This observation prompted us to consider the possibility that crystal packing contacts might have affected the A-loop conformation in these molecular brake mutant structures. Indeed, overlay of the crystal structure of unphosphorylated WT FGFR2K (PDB ID: 2PSQ [*Chen et al., 2007*]) onto those of unphosphorylated autoinhibited FGFR1K (PDB ID: 1FGK [*Mohammadi et al., 1996*]) and FGFR4K (PDB ID: 4QQT [*Huang et al., 2015*]), phosphorylated active FGFR1K (PDB ID: 3GQI [*Bae et al., 2009*]) and FGFR2K (PDB ID: 2PVF [*Chen et al., 2007*]), and mutationally activated FGFR3K[K650E] (PDB ID: 4K33 [*Huang et al., 2013*]) shows that the unphosphorylated WT FGFR2K is not in a genuine inhibited state. First, the location of the αC helix in the unphosphorylated FGFR2K crystal structure is very similar to that in the phosphorylated active FGFR1K and FGFR2K and mutationally activated FGFR3K[K650E] (*Figure 2A*); second, the conformation of the first seven residues of the A-loop (DFGLARD) in the unphosphorylated FGFR2K matches those of the phosphorylated activated FGFR1K and FGFR2K and the mutationally activated FGFR3K[K650E] including $\beta$9 strand formation, which is a characteristic feature of the activated kinases (*Figure 2B*); third, the substrate tyrosine binding site in the unphosphorylated FGFR2K is fully accessible which is in stark contrast to that in the unphosphorylated FGFR1K (PDB ID: 3KY2 [*Bae et al., 2010*]) and FGFR4K (PDB ID: 4QQT [*Huang et al., 2015*]) where a prominent salt-bridge between an FGFR-invariant arginine from the A-loop and the catalytic base occludes the active site, which we have termed the A-loop plug (*Figure 2C*). Notably in the crystal structures of the unphosphorylated FGFR2K (2PSQ) and its molecular brake mutants (2Q0B, 2PWL, 2PZ5, 2PY3, 2PZR), a tightly bound sulfate ion occupies a similar location as the phosphate moiety of the phosphorylated A-loop tyrosine 657 (pTyr-657) in the phosphorylated active FGFR2K structure. This sulfate ion makes hydrogen bonds with the invariant arginine from the DFGLARD motif reminiscent of pTyr-657 in the phosphorylated active FGFR2K. Hence, it appears that the partially active kinase conformation seen in the crystal structures of the unphosphorylated WT FGFR2K (2PSQ) and its molecular brake mutants have been influenced by crystal packing and the presence of the sulfate ion trapped in the A-loop.

In light of this ambiguity, we tried crystallizing the unphosphorylated FGFR2K in different space groups in hopes of eliminating the bias introduced by the lattice contacts on the A-loop conformation in *2PSQ*. However, these attempts were to no avail. We would like to mention that other groups have faced the same problem with FGFR2K crystals (*Eathiraj et al., 2011*). Evidently, strong lattice contact forces prevail and conceal a truly autoinhibited conformation of the A-loop in the unphosphorylated form of FGFR2K. Hence, we decided to use NMR spectroscopy to assess the conformation of the unphosphorylated FGFR2K in solution (i.e. free of crystal lattice contacts). Guided by crystallographic data, two sets of NMR experiments were designed to obtain proximity information between the A-loop and the C-lobe in the unphosphorylated form of FGFR2K in solution. According to the crystal structures, the A-loop tyrosines in unphosphorylated FGFR1K (and FGFR4K) are tethered to the C-lobe, whereas in the unphosphorylated FGFR2K the A-loop including the tyrosines are

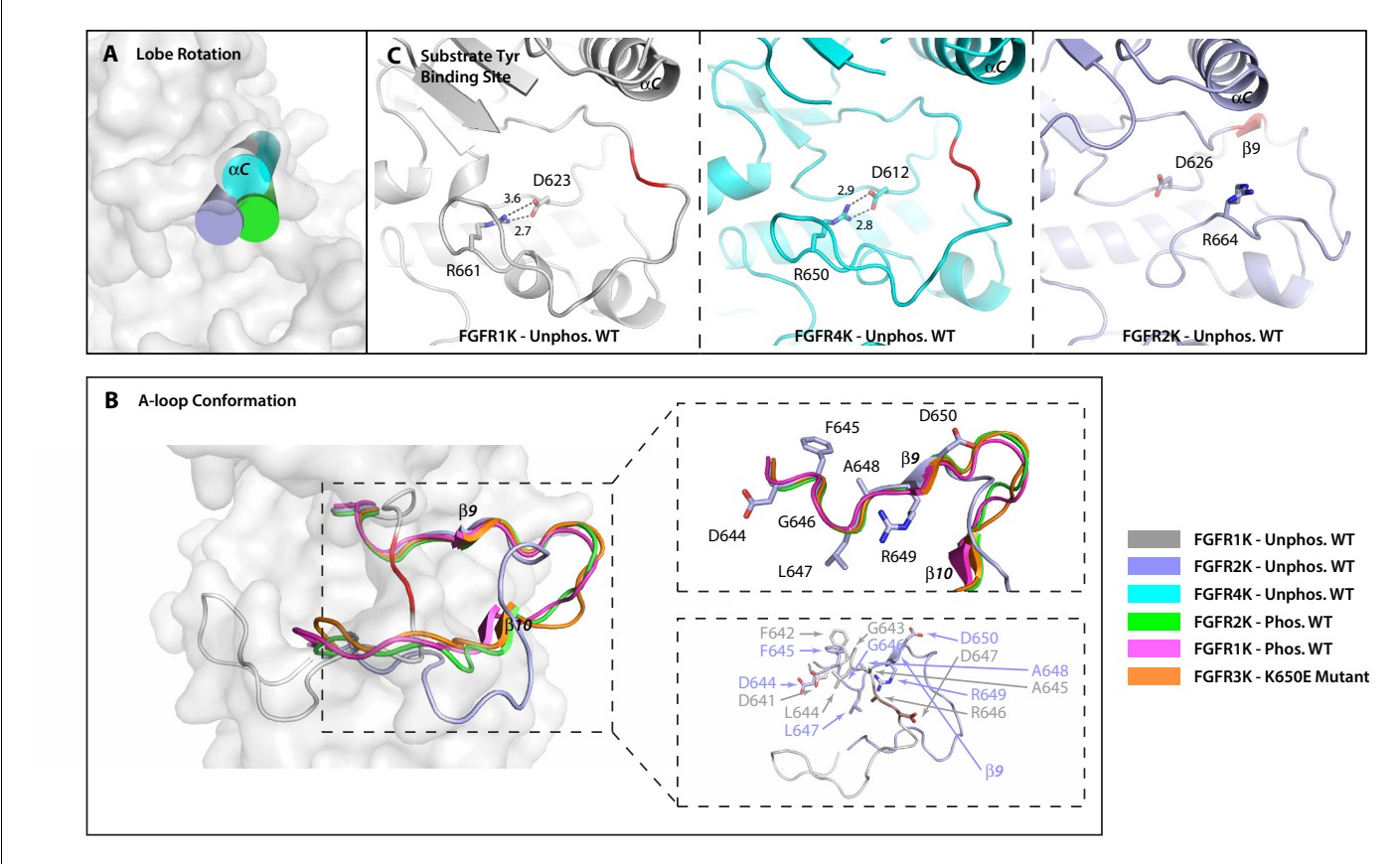

**Figure 2.** The crystal structure of unphosphorylated WT FGFR2K is in a partially active conformation. (**A**) The orientation of the αC helix (rendered as a cylinder) in the crystal structure of unphosphorylated WT FGFR2K is similar to those seen in crystal structures of the A-loop phosphorylated FGFR1K and FGFR2K. (**B**) The conformation of first seven residues of the A-loop (DFGLARD) of the unphosphorylated WT FGFR2K resembles those of phosphorylated activated FGFR1K and FGFR2K. Note that as in these latter phosphorylated activated FGFRK structures, the unphosphorylated FGFR2K features the β9 strand which pairs with the catalytic loop. (**C**) The substrate tyrosine binding site in the unphosphorylated FGFR2K is fully accessible which is in stark contrast to that in the unphosphorylated FGFR1K and FGFR4K in which a prominent salt-bridge between an FGFR-invariant arginine (R661 in FGFR1K, R664 in FGFR2K and R655 in FGFR4K) and the catalytic base (D623 in FGFR1K, D626 in FGFR2K and D617 in FGFR4K) blocks the active site. Hydrogen bonds are shown as dashed lines with the distances given in Å. The A-loop section in the unphosphorylated FGFR1K and FGFR2K that forms a β9 strand upon A-loop phosphorylation is highlighted in red. Note that the β9 strand is already present in the unphosphorylated FGFR2K implying that this structure is in a partially active state.

The following figure supplement is available for figure 2:

**Figure supplement 1.** PRE experiments and chemical shift analysis provide evidence of an FGFR1K-like autoinhibited A-loop conformation for FGFR2K.

stabilized away from the C-lobe by the favorable lattice contacts. Notably, in all the activated forms of FGFRKs, the A-loop adopts a common open conformation and makes few contacts with the C-lobe. Hence, if the A-loop tyrosines in the unphosphorylated FGFR2K were positioned similarly as in FGFR1K (and FGFR4K) then upon A-loop phosphorylation, the residues in the C-lobe of FGFR2K would be expected to experience chemical shift perturbations. Plotting of the chemical shift differences between the unphosphorylated and monophosphorylated (pY657) activated FGFR2K onto the crystal structures of unphosphorylated FGFR1K (3KY2) and FGFR2K (2PSQ) shows significant perturbations for several residues in the C-lobe including the αEF helix (L675), αF helix (V691), and αG helix (I707, V709, E711, L712, F713, K717, E718, and M722) (*Figure 2—figure supplement 1A*). These NMR results cannot be reconciled with the observed conformation of the A-loop in the FGFR2K structure (2PSQ) as the A-loop tyrosines in this structure do not interact with these perturbed residues in the C-lobe. Hence, the chemical shift mapping analysis implies that the true

autoinhibited conformation of the A-loop in FGFR2K is more similar to that observed in crystal structures of FGFR1K (3KY2) and FGFR4K (4QQT) than in the unphosphorylated FGFR2K structure. In addition to mapping chemical shift differences, we also carried out paramagnetic relaxation enhancement (PRE) experiments to directly probe the distance between the A-loop tyrosines and the C-lobe of the enzyme. PRE effects are strong up to a distance of ~20 Å for nitroxide spin labels and start to dissipate at distances longer than ~25 Å. Since PRE experiments require a surface-exposed free cysteine for covalently attaching a paramagnetic spin label onto the protein, we mutated I707 to cysteine. Located in the αG helix of FGFR2K, I707 is ideally suited to discern the A-loop conformational difference between unphosphorylated FGFR2K (2PSQ) and FGFR1K/FGFR4K (3KY2/4QQT) structures (*Figure 2—figure supplement 1B*). In the unphosphorylated FGFR2K structure (2PSQ), the Cγ of I707 and the nitrogen atom of Y656 are ~27 Å apart, whereas the corresponding distance between V704 and Y653 in FGFR1K (3KY2) is much shorter (~15 Å). Following selective enrichment with $^{15}$N tyrosine, the I707C FGFR2K mutant was covalently reacted with MTSL nitroxide (1-oxyl-2,2,5,5-tetramethyl-3-pyrroline-3-methyl)-methanethiosulfonate), and $^{1}$H/$^{15}$N HSQC spectra were then acquired on both the oxidized (paramagnetic) and reduced forms (diamagnetic; treated with ascorbic acid) of the MTSL labeled I707C mutant (*Figure 2—figure supplement 1B*). The intensity retentions were calculated by dividing the peak heights for the oxidized and reduced sample spectra, and following the approach from Battiste and Wagner (*Battiste and Wagner, 2000*), these values were converted into the distances shown in *Figure 2—figure supplement 1B*. Both Y656 and Y657 were found to be ~19 Å away from the MTSL-labeled C707, which is in better agreement with the A-loop conformation in the FGFR1K structure (3KY2) than the A-loop conformation in FGFR2K (2PSQ). While this distance is slightly longer than that measured in the FGFR1K crystal structure, it is within the typical range of experimental uncertainty (±4 Å) (*Battiste and Wagner, 2000*) stemming from the packing of MTSL against the protein. Hence, as with the chemical shift data, our PRE results also imply that the A-loop conformation in the unphosphorylated FGFR2K structure (2PSQ) is not a genuinely inhibited conformation, and that in solution the A-loop of FGFR2K most likely adopts a similar conformation as that displayed by the A-loops of unphosphorylated FGFR1K (3KY2) and FGFR4K (4QQT) in the crystal.

To resolve the ambiguity surrounding the A-loop conformation of the molecular brake mutants, we also retried crystallizing FGFR2K mutants bearing single and double mutations of the molecular brake constituents. Fortunately, we succeeded in obtaining the crystal structure of the FGFR2K$^{E565A/N549H}$ double mutant in a different space group than we previously solved for WT FGFR2K (*Table 1*). Similar to crystal structures of single mutants of the molecular brake, the FGFR2K$^{E565A/N549H}$ structure showed a disengaged molecular brake (*Figure 3A*). However, unlike these previous structures the A-loop in the FGFR2K$^{E565A/N549H}$ structure adopted a catalytically active conformation with striking resemblance to that of A-loop phosphorylated FGFR2K (*Figure 3B*). Taken together with the structures of A-loop mutants, these new structural data further support the existence of long-range allostery between the molecular brake and A-loop that are separated by ~25 Å.

## Gain-of-function mutations expose two novel regulatory sites that serve as intermediaries in the allosteric communication between the A-loop and the molecular brake

Next, we aimed to delineate the trajectory of the allosteric connectivity between the A-loop and molecular brake. A clue into the hotspots of this allosteric path was provided by the analysis of five gain-of-function mutations from the FGFR2K isoform that map to the aαC helix (M537I), αC–β4 loop in the close vicinity of molecular brake (I547V), αE helix (L617M/F), and A-loop (D650V) (*Table 2*). The M537I and L617M mutations confer resistance to the anticancer drug ponatinib (*Byron et al., 2013*), and M537I also corresponds to the M528I mutation in FGFR3 that was recently identified in patients with proportionate short stature (*Kant et al., 2015*). I547V is a somatic mutation detected in endometrial carcinoma patients (*Pollock et al., 2007*) and corresponds to the gain-of-function I538V mutation in FGFR3, which is responsible for hypochondrolasia syndrome (*Grigelionienè et al., 1998*). The L617F mutation is a naturally occurring gain-of-function mutation found in patients with Crouzon craniosynostosis syndrome (*Goos et al., 2015*; *Suh et al., 2014*). Lastly, the D650V mutation, which is not an authentic pathogenic FGFR mutation, corresponds to D816V, D842V and D835V mutations of KIT, PDGFR and FLT3, respectively, that frequently occurs in human Systemic mastocytosis (*Lim et al., 2009*; *Garcia-Montero et al., 2006*), gasterointestinal stromal (GIST)

**Table 1.** X-ray data collection and refinement statistics.

| Construct | E565A/N549H | E565A/K659M | D650V | E565A/D650V |
|---|---|---|---|---|
| *Data Collection* | | | | |
| Resolution (Å) | 50–2.90 (2.95–2.9) | 50–2.05 (2.09–2.05) | 50–1.86 (1.93–1.86) | 50–2.35 (2.39–2.35) |
| Space group | $P2_12_12_1$ | $P4_322$ | $P2_12_12_1$ | $P2_12_12_1$ |
| Unit Cell Parameters (Å, °) | a = 67.334<br>b = 78.557<br>c = 116.546<br>α = 90.00<br>β = 90.00<br>γ = 90.00 | a = 73.937<br>b = 73.937<br>c = 311.359<br>α = 90.00<br>β = 90.00<br>γ = 90.00 | a = 67.215<br>b = 78.786<br>c = 116.544<br>α = 90.00<br>β = 90.00<br>γ = 90.00 | a = 66.973<br>b = 77.860<br>c = 115.809<br>α = 90.00<br>β = 90.00<br>γ = 90.00 |
| Content of the asymmetric unit | 2 | 2 | 2 | 2 |
| Measured reflections (#) | 197265 | 501457 | 648645 | 368127 |
| Unique Reflections (#) | 14079 | 55036 | 46452 | 25875 |
| Data redundancy | 14.0 (12.6) | 9.1 (4.0) | 14.0 (11.0) | 14.2 (11.6) |
| Data completeness (%) | 100 (100) | 98.8 (94.7) | 87.9 (57.6) | 99.8 (99.8) |
| $R_{sym}$ (%) | 14.8 (36.3) | 3.3 (8.6) | 4.8 (18.2) | 8.7 (25.5) |
| I/sig | 26.5 (7.2) | 76.9 (22.6) | 75.5 (17.3) | 42.3 (9.5) |
| *Refinement* | | | | |
| R factor/R free | 25.41/31.12 | 18.22/20.67 | 21.90/25.81 | 21.51/27.49 |
| Number of protein atoms | 4330 | 4830 | 4452 | 4570 |
| Number of non-protein/solvent atoms | 74 | 62 | 72 | 84 |
| Number of solvent atoms | 2 | 258 | 182 | 38 |
| Rmsd bond length (Å) | 0.002 | 0.008 | 0.007 | 0.009 |
| Rmsd bond angle (°) | 0.467 | 0.969 | 0.882 | 1.109 |
| *PDB ID* | 5UHN | 5UI0 | 5UGL | 5UGX |

The value in parenthesis refer to the highest resolution shell.

tumors (*Corless et al., 2005*), and acute myeloid leukemia (*Yamamoto et al., 2001*; *Abu-Duhier et al., 2001*). Notably, KIT, PDGFR, and FLT3 all share the same network of hydrogen bonds at the molecular brake as FGFR2K, suggesting that these kinases are also subject to autoinhibition by this regulatory site (*Chen et al., 2007*). Indeed, somatic mutations of the conserved asparagine from the molecular brake triad activate these RTKs in GIST tumors (*Corless et al., 2005*; *Dutt et al., 2008*; *Kinoshita et al., 2007*).

Analysis of the I547V and L617M/F mutations in the context of autoinhibited and activated FGFRK crystal structures revealed a previously unrecognized cluster of hydrophobic interactions centering on the phenylalanine from the RTK-invariant DFG motif (*Figure 4C,E and G*). This hydrophobic cluster, which we have termed the DFG latch, determines the conformation of the N-terminal part of the A-loop and hence the N-lobe rotation. Specifically, in the autoinhibited FGFR1K, FGFR3K, and FGFR4K structures, the conformations of the DFG and adjacent LAR residues are physically incompatible with downward rotation of the αC helix toward the C-lobe, a well-documented mandatory event in kinase activation (*Figure 4A*). The inhibited conformation of the DFGLAR motif is primarily dictated by the DFG latch where the phenylalanine is fixed in its observed inhibitory position through tight hydrophobic interactions with L617 from the αE helix, I553 from the αC-β4 loop, and I541 from the αC helix of FGFR2K (*Figure 4C,E and G*). In the phosphorylated activated FGFRK structures, however, an RTK-invariant salt bridge between the arginine within the DFGLAR motif and the phosphotyrosine in the A-loop is formed. This intraloop salt bridge facilitates backbone hydrogen bonding between this same arginine and the conserved I623 situated in the catalytic loop resulting in the formation of the short β6-β9 sheet (*Figure 4B*). This structural change alters the conformation of the preceding DFG latch causing the $\chi_1$ side chain dihedral angle of F645 to rotate by 120° which places the side chain of F645 in a new location that is now conducive with downward

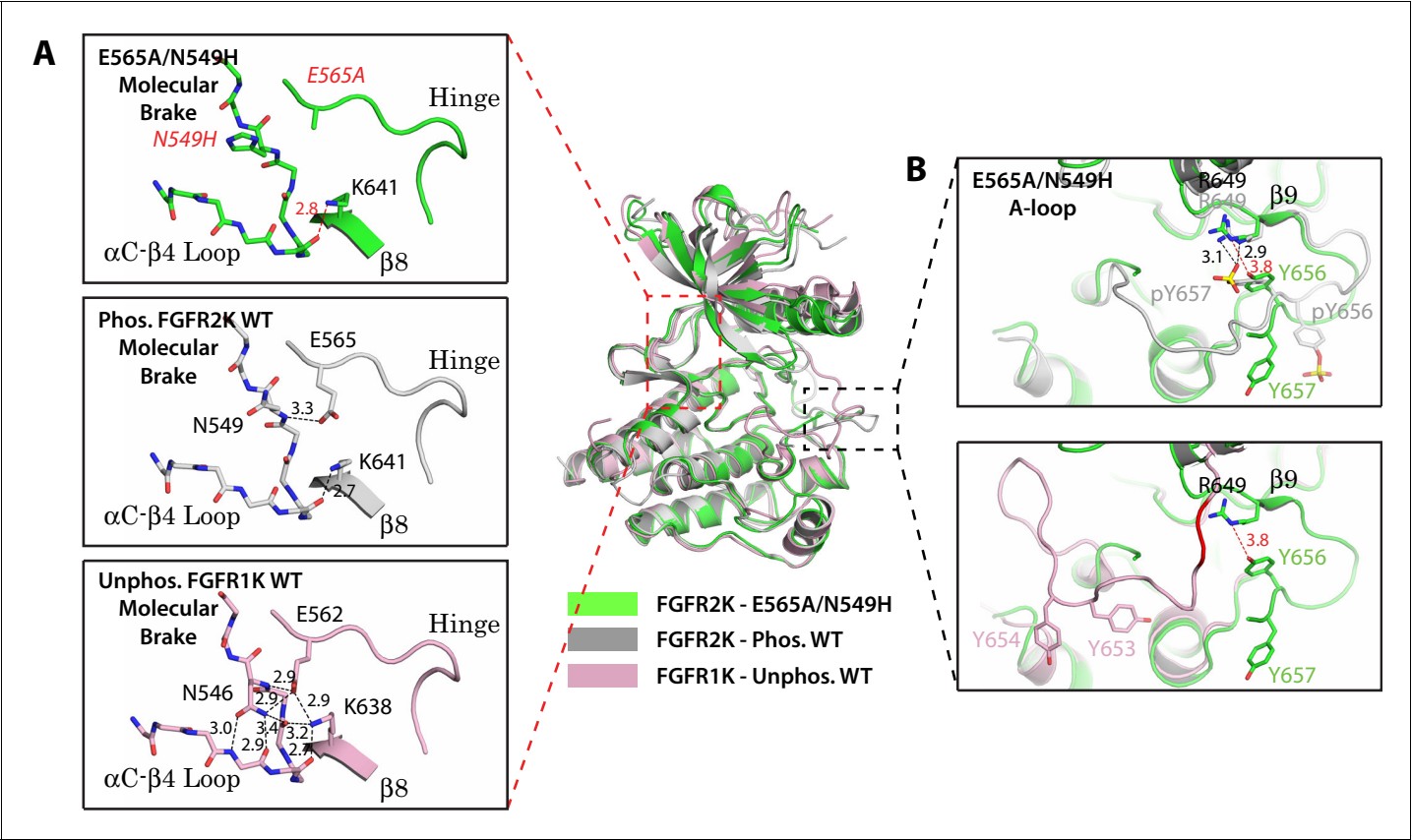

**Figure 3.** Two mutations within the molecular brake disrupt the hydrogen bonding at the molecular brake and drive the A-loop into the active conformation. Crystal structures are displayed for the E565A/N549H double mutant of FGFR2K (in green), phosphorylated activated WT FGFR2K (PDB ID: 2PVF [*Chen et al., 2007*], in grey), and unphosphorylated autoinhibited WT FGFR1K (PDB ID: 1FGK [*Mohammadi et al., 1996*], in pink). (A) Close-up views of the molecular brake region for each of the three structures. Note that reminiscent of the A-loop phosphorylated activated FGFR2K, the molecular brake is disengaged in the unphosphorylated E565A/N549K structure. The two mutated residues, namely N549H and E565A, are labeled with red letters. (B) Comparison of the A-loop conformations for the three structures. The top panel shows a zoomed-in view of the A-loop conformation for E565A/N549K superimposed onto that of phosphorylated FGFR2K. The bottom panel shows a zoomed-in view of the A-loop conformation for E565A/N549K superimposed onto that of unphosphorylated WT FGFR1K. Backbone residues in unphosphorylated FGFR1K within the A-loop that form the β9 strand after A-loop phosphorylation are highlighted in red. Hydrogen bonds are shown as dashed black (WT structures) and red (mutant structure) lines with distances displayed in Å. Side chains of relevant residues are shown as sticks. Atom colorings are the same as in *Figure 1*.

rotation of the N-lobe (*Figure 4A*). Notably, in the activated kinase, the hydrophobic DFG latch becomes less engaged relative to the autoinhibited kinase as indicated by longer distances between the phenylalanine and the other three constituents of the latch (I541, I547, L617) as well as a decrease in shape complementarity (*Figure 4D,F and I*). According to this structural analysis, the L617M/F and I547V gain-of-function mutations act by weakening the DFG latch that encourages the movement of the N-lobe toward the C-lobe, which under physiological conditions would be accomplished by A-loop tyrosine phosphorylation. In addition, we postulate that the weakened hydrophobic contact between F645 of the DFG motif and I547, which is situated on the same αC-β4 loop as N549, perturbs the hydrogen bonding network at the molecular brake resulting in its disengagement. It should be noted that the DFG conformation seen in the activated FGFRK structures is distinct from the so-called 'DFG-out' conformation induced by binding of type II ATP competitive inhibitors such as Ponatinib and Gleevec (*Schindler et al., 2000*).

Structural analysis of the M537I drug-resistant mutation and the adopted D650V pathogenic mutation exposed a second regulatory site, termed the αC tether. In autoinhibited FGFRK structures, the conserved aspartic acid (D650 in FGFR2K) does not play any role in stabilizing the autoinhibited conformation of the A-loop (*Figure 5A and B*). However, in activated kinase structures, it

**Table 2.** Summary of FGFR kinase single and double mutations used in the article.

| FGFR2K single mutation | Location |
|---|---|
| M537I | αC tether |
| M537A | αC tether |
| M540A | αC tether |
| I547V (I544V in FGFR1K) | DFG latch |
| L617M | DFG latch |
| L617F | DFG latch |
| L617V (L614V in FGFR1K) | DFG latch |
| N549K | molecular brake |
| N549H | molecular brake |
| N549T | molecular brake |
| E565A | molecular brake |
| K641R | molecular brake |
| D650A | αC tether |
| D650L | αC tether |
| D650V | αC tether |
| D650I | αC tether |
| D650G | aC tether |
| K659E (K650E in FGFR3K) | A-loop |
| K659M | A-loop |
| **FGFR2K Double Mutations** | **Locations** |
| N549H/E565A | molecular brake / molecular brake |
| K659M/E565A | A-loop / molecular brake |
| D650V/E565A | αC tether / molecular brake |
| M537I/E565A | αC tether / molecular brake |
| D650V/M537I | αC tether / αC tether |
| I547V/M537I | DFG latch / αC tether |
| L617M/D650V | DFG latch / αC tether |
| I547V/E565A | DFG latch / molecular brake |
| L617M/E565A | DFG latch / molecular brake |

makes van der Waals contacts with two conserved methionines (M537 and M540 in FGFR2K) at the C-terminal end of the αC helix (*Figure 5C*). These van der Waals/hydrophobic contacts cooperate with the aforementioned arginine-phosphotyrosine salt bridge to further promote the $\beta9$-$\beta6$ strand formation (*Figure 5C*), thus encouraging movement of the phenylalanine from the DFG motif to facilitate N-lobe rotation. The structural changes at the αC tether and DFG latch results in breakage of hydrogen bonds at the A-loop plug, which frees up the substrate-binding site. Based on this structural analysis, D650V or M537I mutations in FGFR2K may accentuate the hydrophobic contacts between the A-loop and the aαC helix to facilitate N-lobe rotation and hence kinase activation. Thus, according to our four-site allosteric model, phosphorylation on the A-loop tyrosine triggers a series of concerted allosteric motions to couple all four sites thereby conferring catalytic competency on FGFRKs (see model in *Figure 6*). It should be stressed that the DFG-latch and αC tether terms are not synonymous to the canonical DFG motif and aαC helix terminology. Rather the DFG-latch and αC tether describe two previously unrecognized clusters of interacting residues that are involved in modulating kinase dynamics between the autoinhibited and activated forms of the enzyme. Neither of these clusters were previously known in the RTK field, which were discovered by studying the effects of naturally occurring gain-of-function mutations in FGFRKs.

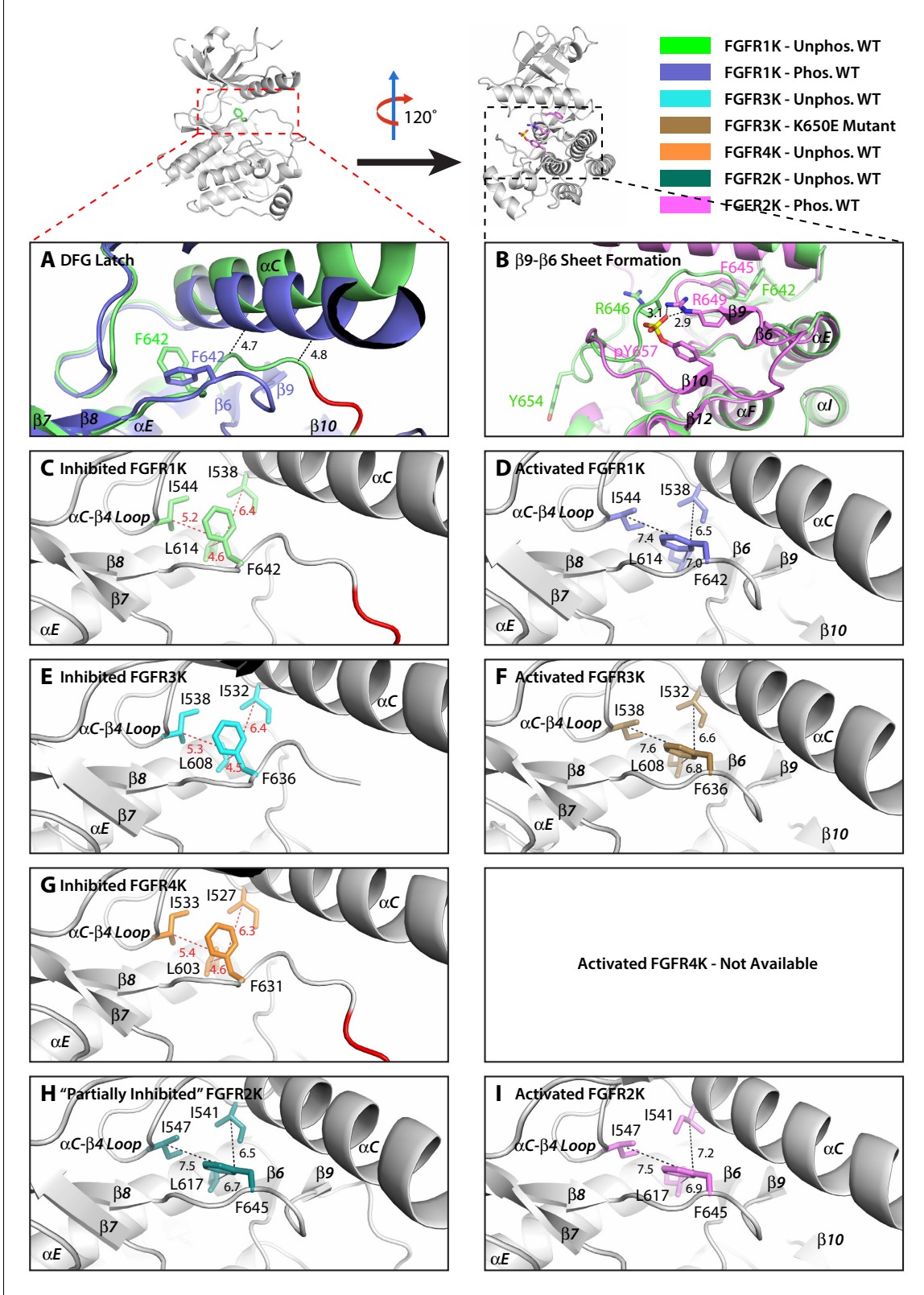

**Figure 4.** The hydrophobic DFG latch in FGFR kinases restrains N-lobe rotation necessary for kinase activation. (**A**) Comparison of the orientation of αC helix and the conformation of the DFGLAR motif at the start of the A-loop between the unphosphorylated autoinhibited WT FGFR1K (PDB ID: 1FGK [*Mohammadi et al., 1996*], in green) and A-loop tyrosine phosphorylated WT FGFR1K (PDB ID: 3GQI [*Bae et al., 2009*], in blue) structures. The side chains of the phenylalanine of the DFG motif are rendered in sticks. The dashed lines are used to illustrate the steric clashes that prevent the αC

*Figure 4 continued on next page*

*Figure 4 continued*

helix from rotating downward in the unphosphorylated autoinhibited FGFR1K structure. The portion of the A-loop in the unphosphorylated WT FGFR1K that becomes β9 strand upon A-loop tyrosine phosphorylation is highlighted in red. (B) Two close-range hydrogen bonds, depicted by dashed lines, between the phosphate moiety of the phosphorylated A-loop tyrosine and conserved arginine (R649 in FGFR1K) stabilize the β9-β6 sheet between the A-loop and the catalytic loop. (C–I) Zoomed-in views of the DFG latch environments observed in the crystal structures of the unphosphorylated WT FGFR1K (PDB ID: 1FGK [*Mohammadi et al., 1996*], in green), phosphorylated WT FGFR1K (PDB ID: 3GQI [*Bae et al., 2009*], in blue), unphosphorylated WT FGFR3K (Chen unpublished, in cyan), K650E gain-of-function mutant of FGFR3K (PDB ID: 4K33 [*Huang et al., 2013*], in brown), unphosphorylated FGFR4K (PDB ID: 4QQT [*Huang et al., 2015*], in orange), unphosphorylated WT FGFR2K (PDB ID: 2PSQ [*Chen et al., 2007*], in dark green), and phosphorylated WT FGFR2K (PDB ID 2PVF [*Chen et al., 2007*], in purple). Distances between the DFG phenylalanine and neighboring leucine and isoleucine residues are indicated by dashed red (autoinhibited kinase) or black (activated kinase) lines with the distance given in Å. Atom colorings are the same as in *Figure 1*.

## Functional and X-ray crystallographic evidence for the role of the DFG latch and αC tether in mediating allostery between the A-loop and the molecular brake

To functionally validate the role of the DFG latch in our allosteric activation model, we studied the impact of weakening hydrophobic contacts that fix the phenylalanine in its observed autoinhibitory orientation. To this end, the activities of I547V, L617M, L617F, L617V single FGFR2K mutants were compared with that of the WT FGFR2K using an in vitro substrate phosphorylation assay coupled with mass spectrometry (*Chen et al., 2013*). As shown in *Figure 7A*, all the mutants exhibited elevated kinase activity compared to WT, which supports our hypothesis that the DFG latch suppresses FGFRK activity. The fact that both I547V mutation in FGFR2K and its analogous I538V mutation in FGFR3K lead to gain-of-function and cause skeletal disorders and cancer (*Pollock et al., 2007*; *Grigelionienė et al., 1998*) strongly suggests that the DFG latch is applicable to all four FGFR isoforms. Indeed, mutations of the corresponding isoleucine and leucine residues of the DFG latch in FGFR1K also caused activation (*Figure 7A* inset).

To test the role of the αC tether in regulating FGFRK activity, we analyzed the impact of strengthening and weakening hydrophobic contacts at the αC tether site. To strengthen the tether, the following single-site FGFR2K mutants were engineered: D650A, D650L, D650V, D650I, and M537I. To weaken the hydrophobic tether, single-site mutations including D650G, M537A and M540A were introduced into FGFR2K. Relative to the unphosphorylated WT FGFR2K, the D650A, D650L, D650V, D650I, and M537I mutants were 3- to 19-fold more active (*Figure 7B*), whereas the D650G, M537A, and M540A mutants were less active (*Figure 7B* inset). These data provide strong support for the role of the αC tether in regulating FGFRK activity.

To provide direct structural evidence for the participation of the αC tether in the allosteric communication network within FGFRK, we determined the crystal structure of the D650V FGFR2K mutant (*Table 1*). According to our model, the D650V mutation should drive the kinase toward the active conformation by fortifying the hydrophobic contacts at the αC tether. Consistent with our prediction, D650V crystallized in a fully activated conformation displaying a disengaged molecular brake and an activated A-loop conformation (*Figure 8A,D and E*). Additional structural analysis showed that the mutated V650 residue makes strong hydrophobic contacts with M537 and M540 between the αC helix and the A-loop thereby facilitating the rotation of the αC helix toward the C-lobe (*Figure 8C*). As expected, the DFG latch is rearranged in a non-inhibitory conformation to facilitate N-lobe rotation (*Figure 8B*), which in turn leads to the disengagement of the molecular brake (*Figure 8A*). Collectively, the X-ray structures and biochemical data support the hypothesis that the hydrophobic DFG latch and αC tether serve as intermediaries in transmitting reciprocal conformational changes between the molecular brake and the A-loop and active site.

## NMR chemical shift perturbation and CPMG experiments support the existence of allosteric connectivity between the molecular brake, DFG latch, αC tether, and A-loop plug

We next applied NMR spectroscopy to validate our proposed allosteric model in solution using the FGFR2K isoform. To begin, we acquired $^1$H/$^{15}$N TROSY-HSQC spectra of isotopically enriched samples of unphosphorylated WT and seven gain-of-function mutants namely, M537I, I547V, N549T,

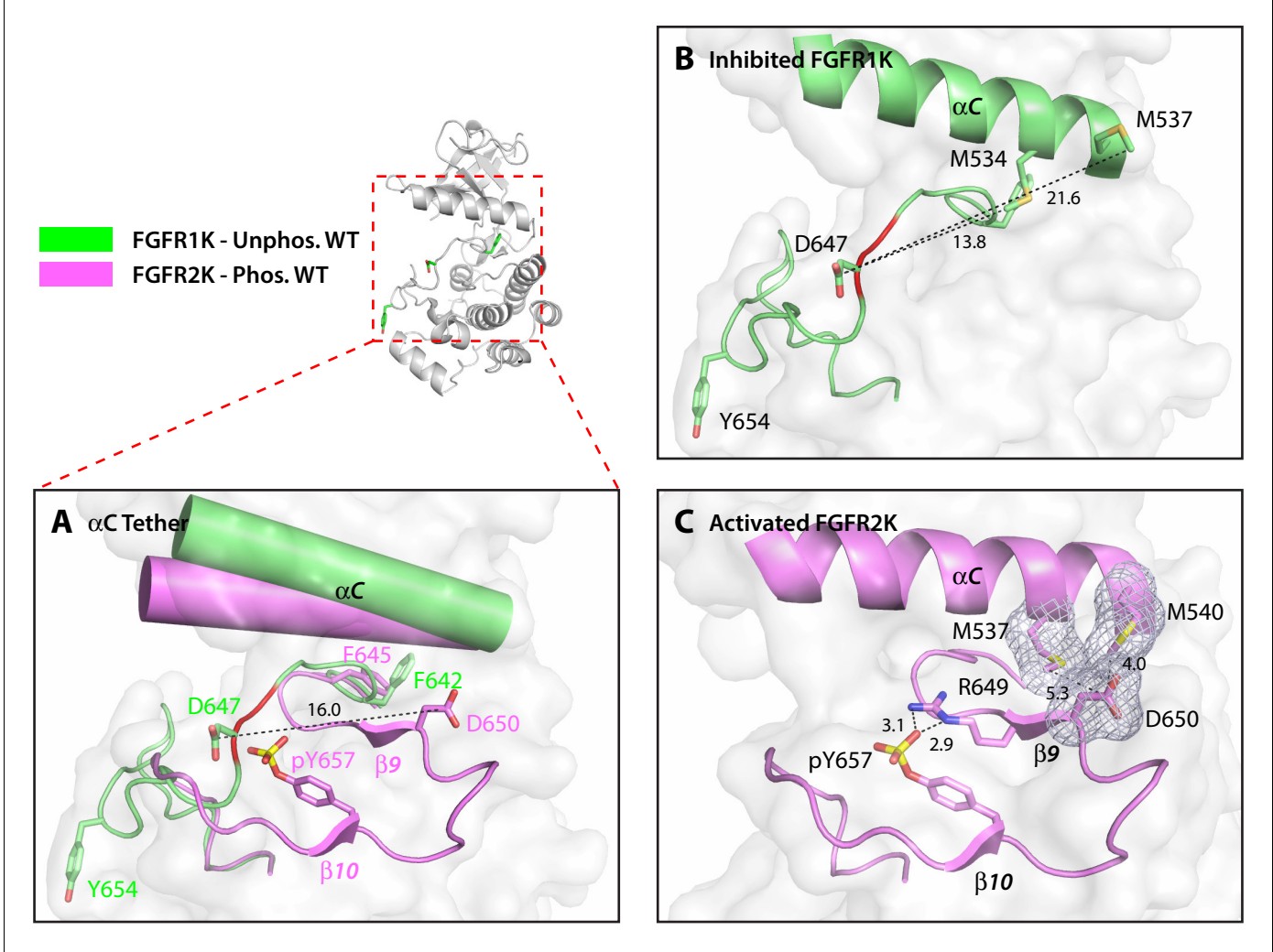

**Figure 5.** The hydrophobic αC tether in FGFR kinase facilitates N-lobe rotation, a mandatory event in kinase activation. (**A**) Structural comparison of the conformation of the αC helix and A-loop between unphosphorylated WT FGFR1K (PDB ID: 1FGK [*Mohammadi et al., 1996*], in green) and A-loop tyrosine phosphorylated WT FGFR2K (PDB ID 2PVF [*Chen et al., 2007*], in purple). Notably, there are major structural and orientation changes of the αC helix, DFG phenylalanine, and A-loop between the two structures. For example, there is a large change in the location of conserved aspartic acid between the two structures (i.e. D647 in FGFR1K and D650 in FGFR2K). (**B**) As highlighted by the distances in Å, D647 in unphosphorylated WT FGFR1K is distant from the two methionines on the αC helix and therefore the aC tether is 'off'. In red is the section of the A-loop in the unphosphorylated FGFR1K that becomes β9 strand upon A-loop tyrosine phosphorylation. (**C**) In the phosphorylated WT FGFR2K structure, the D650 makes van der Waals contacts with the two methionines from the αC helix and the αC tether is 'on'. Engagement of the aC tether acts in concert with a salt bridge between the A-loop phosphotyrosine and conserved arginine to facilitate β9 strand formation. The hydrophobic interactions are represented by semi-transparent mesh. The distances between the aspartic acid and the two methionines on the αC helix are shown as black dashed lines with the distances given in Å. Hydrogen bonds stabilizing the active A-loop conformation are shown as black dashed lines with distances in Å. Atom colorings are the same as in *Figure 1*.

E565A, K641R, D650V, and K659E. For each mutant, a plot of the chemical shift perturbations as a function of residue is displayed in *Figure 9—figure supplement 1* with representative spectra shown in *Figure 9A* for residues near or within the molecular brake (A567), catalytic loop (I623), DFG latch (G646), and αC tether (G542). Mutation at the DFG latch (I547V) caused chemical shift perturbations for residues within the molecular brake, αC tether, and catalytic loop (*Figure 9A*). Likewise, mutations affecting the αC tether (M537I, D650V) gave small but significant chemical shift changes to G646 of the DFG motif and I623 from the catalytic loop. Notably, the D650V mutation induced a perturbation at A567, which is in the vicinity of the molecular brake, whereas the M537I mutation

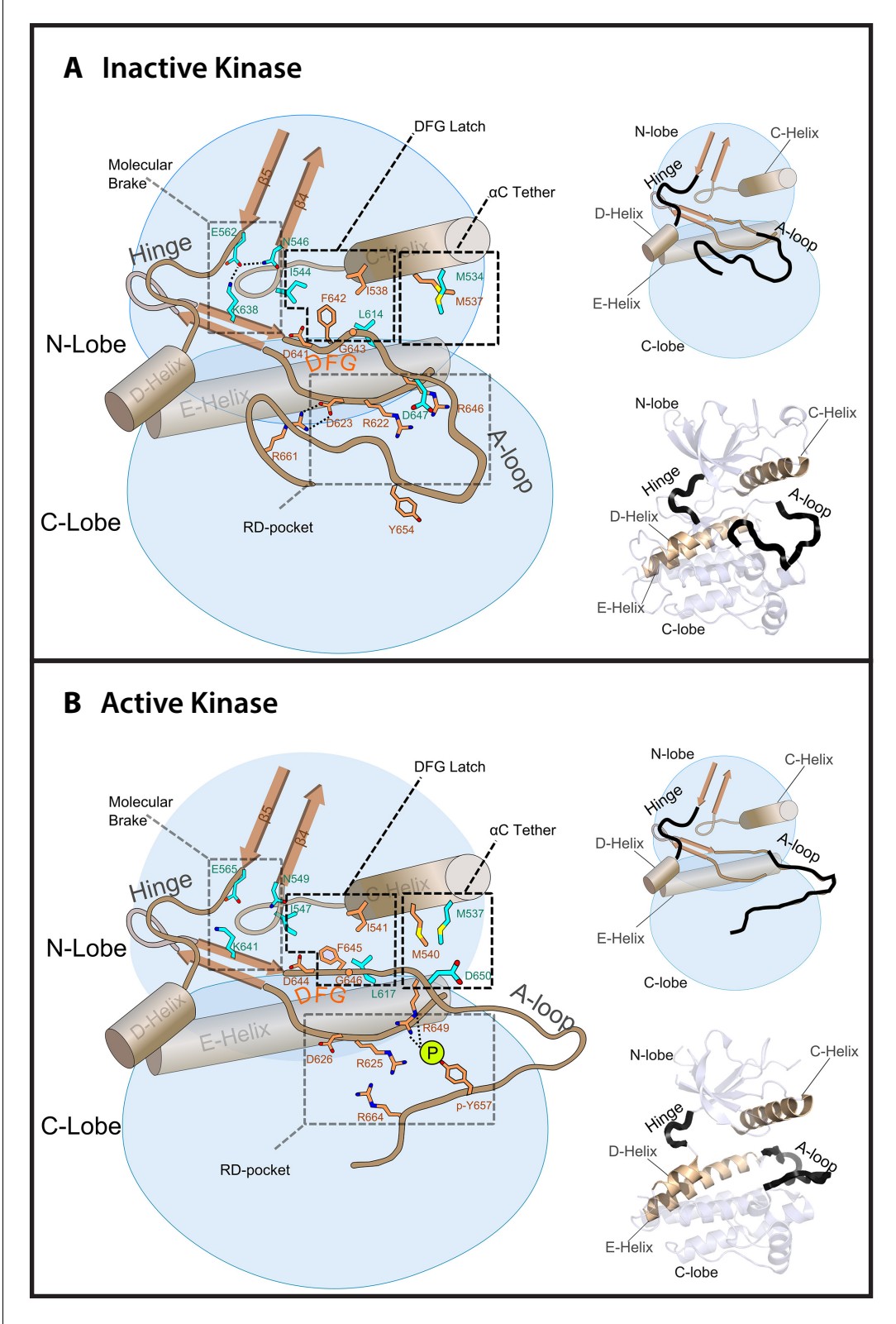

**Figure 6.** Model of the allosteric network for the kinase domain of FGFR. Depiction of the four molecular sites/regions comprising the long-range allostery in FGFRK: (1) molecular brake: an inhibitory network of hydrogen bonds near the kinase hinge; (2) DFG latch: a network of hydrophobic connectivity centered at the DFG motif; (3)

*Figure 6 continued on next page*

*Figure 6 continued*

αC tether: a hydrophobic tether between the αC helix and the A-loop; (4) A-loop plug: a salt bridge between an FGFR-invariant arginine in the A-loop (R661 in FGFR1) and the catalytic base aspartic acid (D623 in FGFR1).

The following figure supplement is available for figure 6:

**Figure supplement 1.** Effect of AMP-PCP binding on kinase activation.

did not significantly perturb this residue. Finally, mutations at the molecular brake (N549T, E565A, K641R) induced significant chemical shift changes to residues at or proximal to the molecular brake, DFG latch, αC tether, and catalytic loop (*Figure 9A*; *Figure 9—figure supplement 1*). Taken together, these backbone chemical shift data show that mutations targeting one of the allosteric sites influence the chemical environment of distal sites and therefore support our proposed four-site long-range allosteric network.

To augment the backbone chemical shift data, we next acquired $^{1}$H/$^{13}$C HMQC data on samples of WT FGFR2K, K659E (A-loop mutant), and E565A (molecular brake mutant) that were selectively $^{13}$C labeled at Ile, Val, and Leu methyl groups. The corresponding methyl group chemical shift perturbation data of E565A and K659E relative to WT are mapped onto the unphosphorylated FGFR1K crystal structure (*Figure 9B*; PDB ID: 3KY2). The E565A mutant incurred major chemical shift changes at residues both proximal to the molecular brake (I548, L551, V564) and distal from the molecular brake at the DFG motif (I541, I547) and A-loop (L647, I651). These data demonstrate that disruption of the hydrogen bonding network of the molecular brake influences the conformation of the A-loop via long-range allostery. Surprisingly, however, the K659E mutant did not exhibit significant backbone or methyl chemical shift perturbations at the molecular brake region (*Figure 9B*). Therefore, to supplement the $^{1}$H/$^{15}$N TROSY-HSQC and $^{1}$H/$^{13}$C HMQC chemical shift perturbation data on the K659E mutant, amide $^{15}$N and methyl $^{13}$C (Ile, Leu, and Val) CPMG relaxation dispersion measurements were carried out (*Loria et al., 1999*; *Korzhnev et al., 2004a*). Note that the CPMG technique is a more sensitive method than chemical shift perturbation analyses to detect sparsely populated conformational states and is routinely used to explore protein dynamics on the μsec-msec timescale (*Mittermaier and Kay, 2009*). Indeed, several residues in the K659E mutant showed significant relaxation dispersions (i.e. $\Delta R_{2,eff} > 2\ s^{-1}$) both at the backbone amide and side chain methyl groups

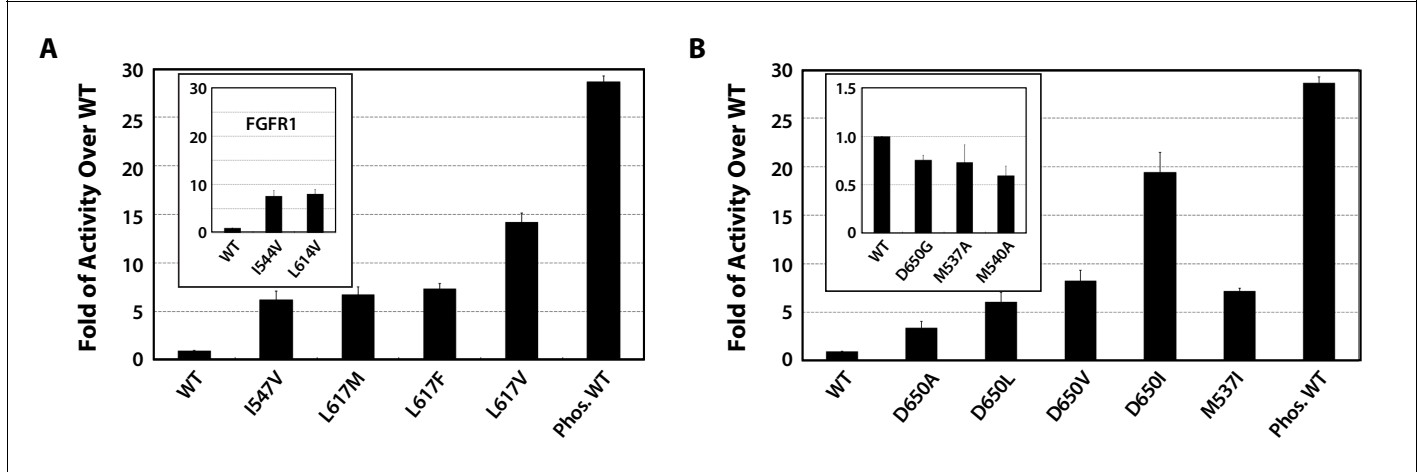

**Figure 7.** Kinase substrate phosphorylation assay confirms the role of the DFG latch and αC tether in FGFR kinase regulation. (**A**) Comparison of the substrate phosphorylation activities of FGFR2K and FGFR1K mutants harboring mutations that weaken the DFG latch with those of the unphosphorylated WT FGFR2K and FGFR1K (inset). The activities are measured at 30 s post reaction and are expressed as fold activity over that of WT FGFR2K or FGFR1K (inset). (**B**) Comparison of the activities FGFR2K mutants harboring mutations that strengthen or weaken the αC tether. The inset shows data for the FGFR2K mutants with a weakened αC tether. Activities are measured at 30 s post reaction and are expressed as fold activity over the activity of the unphosphorylated WT FGFR2K. Error bars represent mean ± SD.

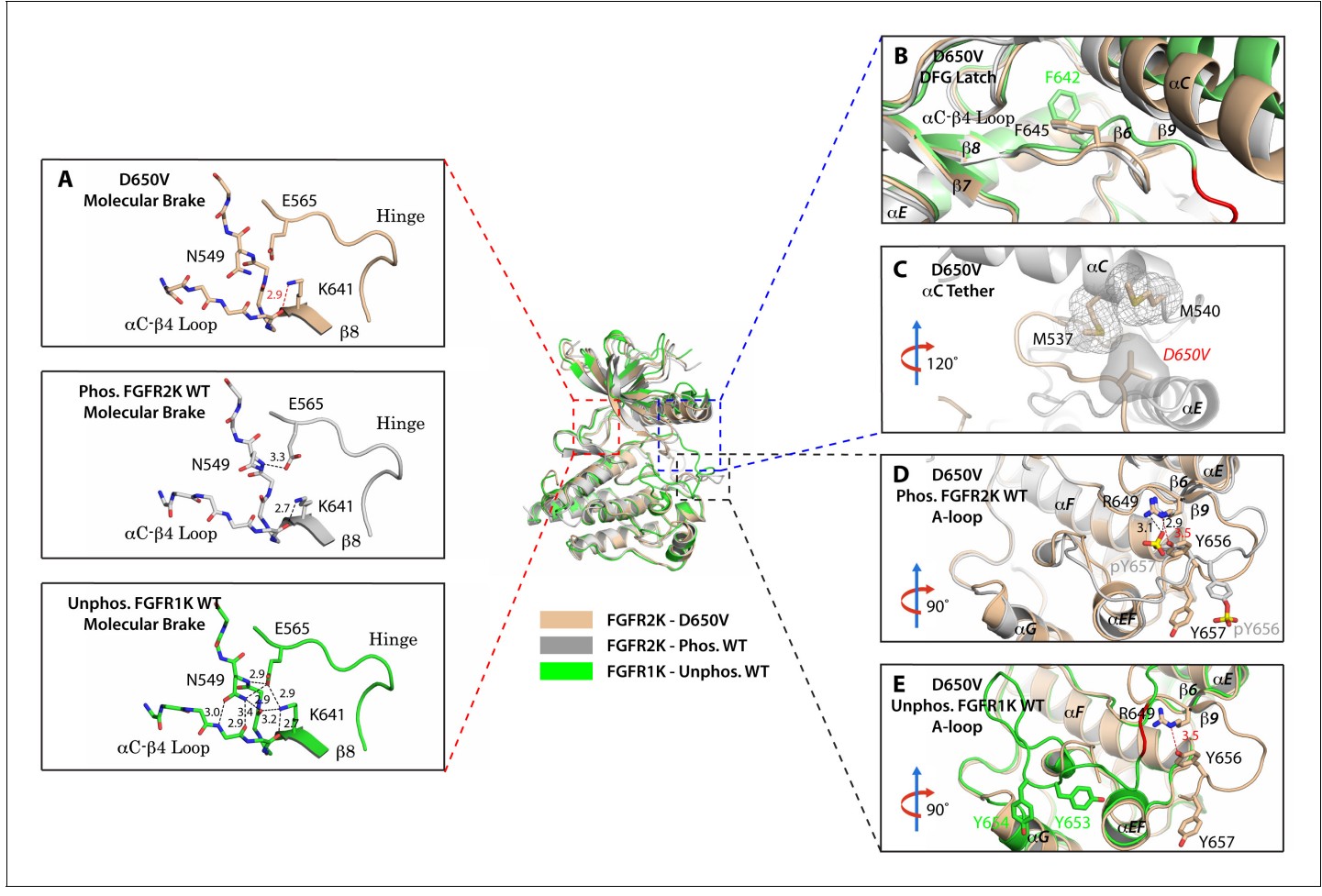

**Figure 8.** The D650V gain-of-function mutation strengthens the αC tether thereby stabilizing the kinase active conformation. Superimposition of the crystal structures of the D650V FGFR2K mutant (in wheat), unphosphorylated autoinhibited FGFR1K (PDB ID: 1FGK [*Mohammadi et al., 1996*], in green), and the A-loop tyrosine phosphorylated WT FGFR2K (PDB ID: 2PVF [*Chen et al., 2007*], in grey). (**A**) Close-up views of the molecular brake regions of each of the structures. Note that the molecular brake is disengaged in the unphosphorylated D650V mutant similar to that of the A-loop phosphorylated activated FGFR2K. (**B**) Close-up view of the DFG latch region showing that the DFG phenylalanine in the D650V mutant adopts the same rotamer position as the corresponding phenylalanine in the A-loop tyrosine phosphorylated WT FGFR2K. (**C**) Close-up view of the αC tether showing enhanced hydrophobic contacts between V650 and methionines in the αC helix. (**D**) Comparison of the A-loop regions of the unphosphorylated D650V mutant and the A-loop tyrosine phosphorylated WT FGFR2K showing that the A-loop in the D650V mutant is primarily in the active conformation. The labels for mutant residues are in red. (**E**) Comparison of the A-loop region of the unphosphorylated D650V mutant and the unphosphorylated inhibited WT FGFR1K showing that the A-loop in the D650V mutant is not in the inhibited conformation. The hydrophobic interactions are represented by semitransparent mesh and solid surface, and the hydrogen bond between pY657 and R649 is shown as a black dashed line with the distance given in Å. Side chains of selected residues are shown as sticks. Atom colorings are the same as in *Figure 1*.

(*Figure 10C–F*). These residues map to the DFG latch (I541, I547, F645), A-loop (L647), catalytic loop (I623, L627), molecular brake (I548, V699), αC tether (M538), and P+1 pocket (I707, V709). To determine the rate of conformational exchange, methyl group CPMG dispersion curves at two magnetic fields corresponding to I548, I623, L627, L647, I707, and V709 were fit in a global fashion using the two-site exchange equation (*Luz and Meiboom, 1963*). Based on this analysis, an exchange rate ($k_{ex}$) of 2000 ± 500 s$^{-1}$ was estimated (*Figure 10F*) indicating that this pathogenic mutant interconverts between the autoinhibited/active states on a μsec-msec timescale (i.e. fast-intermediate exchange). By contrast, CPMG relaxation dispersion measurements on unphosphorylated WT FGFR2K showed very few residues exhibiting relaxation dispersions (*Figure 10A and B*). Based on these observations, we conclude that the unphosphorylated form of FGFR2K is conformationally rigid (i.e. only a small fraction of kinase molecules are in the active state conformation). These NMR

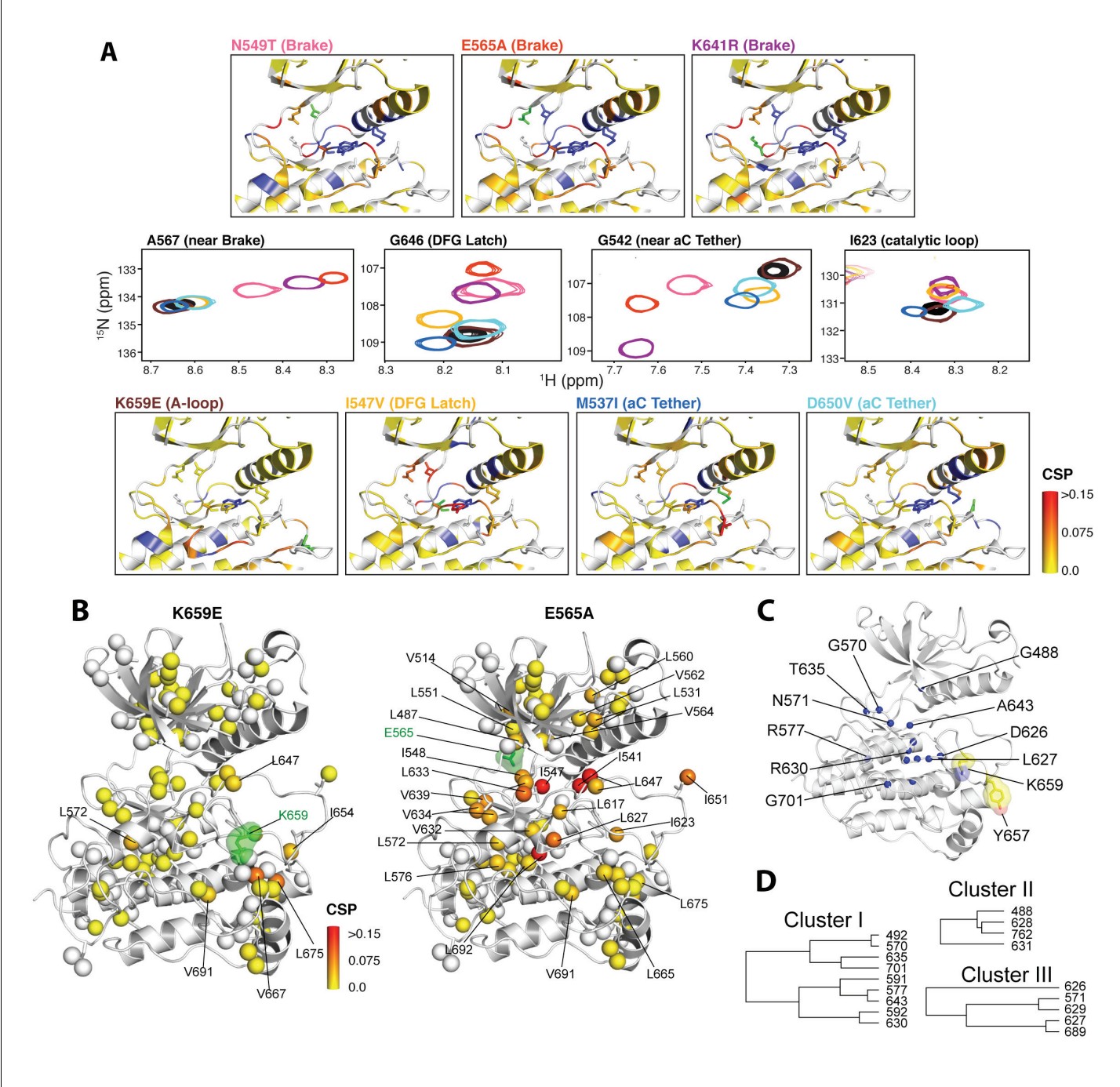

**Figure 9.** NMR chemical shift perturbation and CHESCA analysis for FGFR2K mutations. (**A**) $^1$H/$^{15}$N correlation spectra are shown in the middle four panels for residues A567, G646, G542, and I623 located near the molecular brake, within the DFG latch, near the αC tether, and within the catalytic loop, respectively. The colors of this spectral overlay match to those of the seven mutants listed in the top three and bottom four panels. Perturbations are mapped onto the active WT FGFR2K structure (PDB ID: 2PVF [**Chen et al., 2007**]) with the magnitude of changes reflecting the difference between unphosphorylated WT FGFR2K and the respective activating mutation (red indicates the maximum perturbation, while yellow corresponds with no perturbation). Blue colored regions correspond to residues whose chemical shifts disappeared or shifted beyond detection for the given mutant. The mutated residue in each structure is colored green. (**B**) Ile, Leu, and Val chemical shift perturbations of K659E (left) and E565A (right) relative to those of unphosphorylated WT FGFR2K. The methyl perturbation sites shown in spheres are mapped onto the autoinhibited WT FGFR1K (PDB ID: 3KY2 [**Bae et al., 2010**]); note that the residue numbering convention corresponds to that of FGFR2K. White spheres correspond to residues unassigned or overlapped in the mutants. (**C**) Residues within the allosteric network identified using CHESCA are mapped onto the FGFR1K structure with a sphere at the backbone nitrogen position (PDB ID: 3KY2 [**Bae et al., 2010**]). The chemical shifts from a series of mutations at K659 (T, N, Q, M, E) and WT

*Figure 9 continued on next page*

*Figure 9 continued*
phosphorylated and unphosphorylated FGFR2K were used for the analysis. (D) Phylogenic tree showing the three separate clusters of residues with correlation coefficients |r$_{ij}$| = 0.97. Based on the similar functional network, these three clusters comprise the same allosteric network (*Figure 9—figure supplement 2*).
The following figure supplements are available for figure 9:

**Figure supplement 1.** Chemical shift perturbations for pathogenic mutants of FGFR2K.

**Figure supplement 2.** CHESCA functional network and single-value decomposition analysis.

data are congruent with the X-ray crystal structures showing that the kinase domain of FGFR is subject to a three-level autoinhibition mechanism via the molecular brake, the DFG latch and the blockade of the active site by the A-loop/catalytic loop salt bridge (A-loop plug).

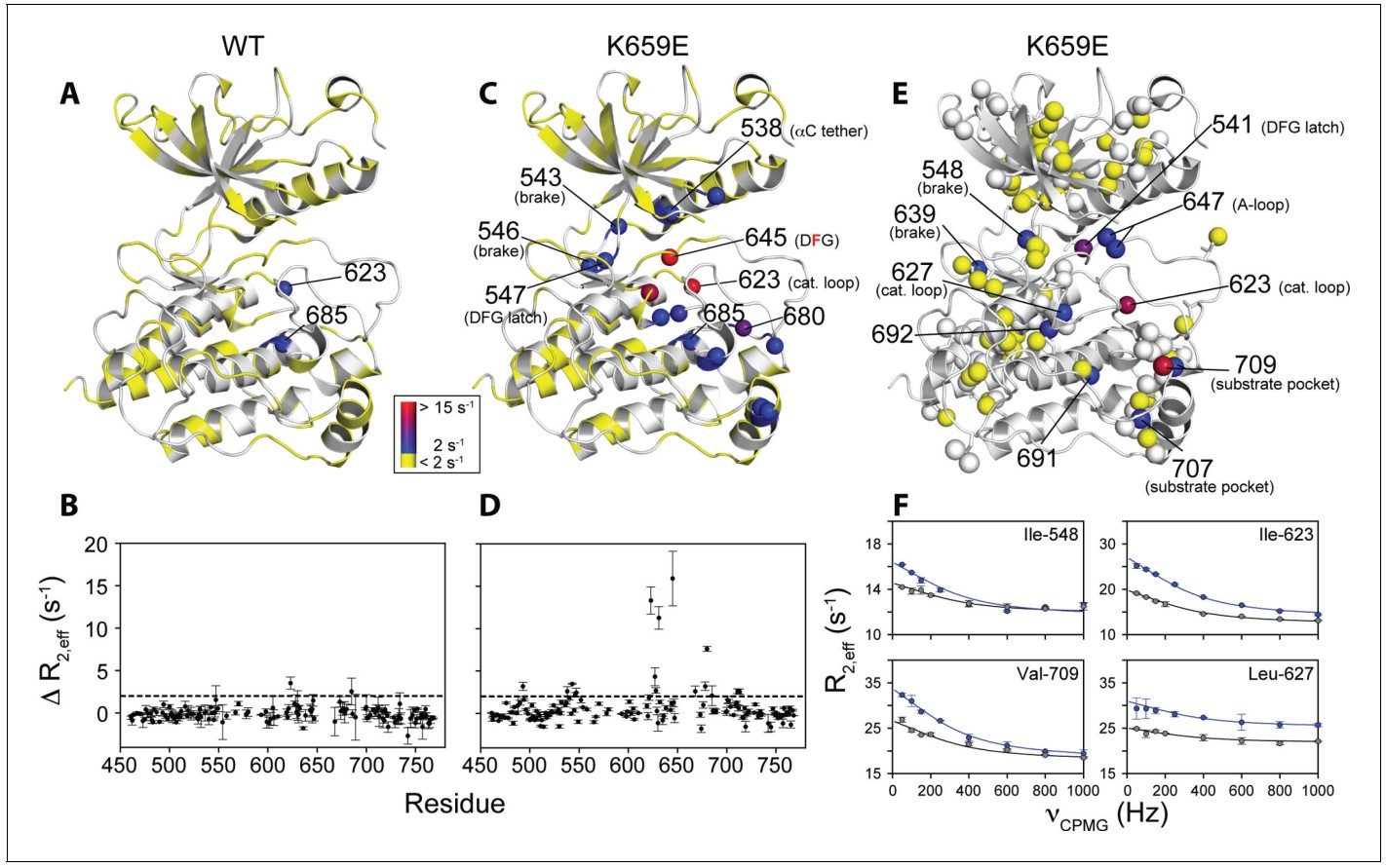

**Figure 10.** Comparison and quantification of microsecond-to-millisecond dynamics for wild-type and K659E FGFR2K. (A–D) $\Delta$R$_{2,eff}$ $^{15}$N CPMG relaxation dispersion values for WT (panel **A**, **B**) and K659E mutant (panel **C**, **D**) plotted onto unphosphorylated WT FGFR1K (PDB ID: 3KY2 *[Bae et al., 2010]*) note that the residue numbering convention corresponds to that of FGFR2K. Significant dispersions (>2 s$^{-1}$) are represented by the dotted line on each $\Delta$R$_{2,eff}$ plot as a function of residue (panels **B** and **D**). All $^{15}$N CPMG data were acquired at a $^1$H frequency of 600 MHz. (E) Results from $^{13}$C multiple quantum CPMG relaxation dispersion experiments acquired at 800 MHz are plotted onto the unphosphorylated WT FGFR1K structure (PDB ID: 3KY2). The $\Delta$R$_{2,eff}$ $^{13}$C values are shown for Ile, Leu, and Val methyl groups (spheres). (F) CPMG dispersion curves for select residues at 800 MHz (blue) and 600 MHz (black). The lines represent a globally fit k$_{ex}$ value equal to 2000 ± 500 s$^{-1}$. For panels **A**, **C**, and **E**, significant dispersions (>2 s$^{-1}$) are represented on a scale from blue to red with dispersions below 2 s$^{-1}$ shown in yellow. White regions (**A** and **C**) or white spheres (**E**) indicate unassigned or unresolved residues.

To further investigate the conformational dynamics of a fully activated FGFRK, we obtained $^1H/^{15}N$ backbone and $^1H/^{13}C$ methyl correlation spectra for FGFR2K monophosphorylated on the A-loop tyrosine Y657. Comparison of these spectra with those of unphosphorylated (inhibited) WT FGFR2K showed a substantial reduction in peak intensities for the DFG motif and catalytic loop in the monophosphorylated activated FGFR2K. These observations reiterate that the DFG motif in the autoinhibited kinase is conformationally rigid (i.e. DFG latch on), whereas in the active form it becomes mobile with conformational fluctuations occurring on the μsec-msec timescale. Notably, CPMG relaxation dispersions experiments performed on K659E also showed significant relaxation dispersions for F645 and the surrounding methyl groups of I541 and I547. These data further support that kinase activation by A-loop phosphorylation or by pathogenic mutations enhances conformational dynamics of the enzyme and leads to a weakening of the DFG latch consistent with our structural analysis. Thus, the observed conformation of F645 side chain seen in the static X-ray structures of activated FGFR kinases represents the thermodynamically most stable orientation of the highly mobile phenyl ring in the active state. Our data are consistent with the previously published NMR data for other activated protein kinases including protein kinase A (*Masterson et al., 2008*) and p38 MAP kinase (*Vogtherr et al., 2006*), which also showed broadening around the DFG motif. Hence, it appears that enhanced conformational dynamics on the μsec-msec timescale for the DFG motif upon activation might be a general feature of all protein kinases.

## CHESCA analyses of the panel of five gain-of-function mutations at K659 in the A-loop provide further support for long-range allostery

To gain further insight into the trajectory of the long-range allosteric network, we applied chemical shift covariance analysis (CHESCA) (*Selvaratnam et al., 2011*) to the backbone chemical shift data of unphosphorylated and A-loop phosphorylated wild-type and unphosphorylated mutated FGFR2Ks carrying a pathogenic mutation at lysine 659 (K659T, K659Q, K659N, K659M, K659E). Specifically, inter-residue pairwise correlations ($|r_{ij}| > 0.97$) were clustered using the complete linkage clustering approach as described previously (*Selvaratnam et al., 2011*; *Boulton et al., 2014*). Three individual clusters of residues (*Figure 9C,D*) were found to each match closely with the relative kinase activities of the K659 mutants (*Figure 9—figure supplement 2*). In addition, the single value decomposition method was used to show that each of these residues within the individual clusters aligned along the same principle component axis (*Figure 9—figure supplement 2*), which further confirmed the existence of a common allosteric network (*Selvaratnam et al., 2011*). In total, 18 residues were found to comprise the same allosteric network shown in *Figure 9C and D*. Mapping these sites onto the autoinhibited FGFR1K structure (PDB ID: 3KY2) illuminated a network of allosterically coupled residues belonging to the catalytic loop (D626 to N631), region surrounding the molecular brake (T635), αF helix (F689), Gly-rich loop (G488, F492), and A-loop (A643). The identification of A643, which is located immediately before D644 of the DFG motif, is harmonious with the crystallographically deduced role of the DFG latch as a conduit in the allostery transmission from the A-loop to the molecular brake. Likewise the identification of D626 and other catalytic loop residues by CHESCA strongly supports a major conformational change of the A-loop between the inhibited and activated structures, which is induced by A-loop tyrosine phosphorylation and mimicked by pathogenic mutations. As mentioned earlier, in the autoinhibited FGFRK structures (e.g., FGFR1K, PDB ID: 3KY2), D623 in FGFR1K corresponding to D626 in FGFR2K engages in a salt bridge with R661 from the A-loop thereby blocking the substrate-binding site (*Figure 2C*). Upon activation this arginine residue disengages from the aspartate and makes a cation-Π interaction with the incoming substrate tyrosine (*Huang et al., 2013*; *Chen et al., 2008*). Taken together, the chemical shift perturbation data and CHESCA analysis support the existence of an allosteric connectivity between the molecular brake and A-loop that is mediated through the DFG latch and αC tether. Importantly, the regions of the kinase identified using CHESCA match to those identified by CPMG experiments, although the latter method detected additional residues involved in the conversion between the activated and autoinhibited conformations not picked up by CHESCA.

## ATP binding is a poor extrinsic effector of the conformational balance between the active and inhibited states

Binding of ATP into the cleft between the N- and C-lobes of the kinase involves residues from the molecular brake and the conserved DFG motif. Notably the amino group of the ATP adenine ring makes a direct hydrogen bond with the backbone carbonyl oxygen of an FGFR-invariant glutamic acid whose side chain is a central mediator of the hydrogen bonding network at the molecular brake. Moreover, the conserved aspartic acid from the tyrosine kinase-invariant DFG motif participates in ATP binding by coordinating a $Mg^{2+}$ ion in concert with the $\alpha$ and $\beta$ phosphate groups of ATP. Based on these structural observations, ATP binding may potentially promote the active state conformation of FGFRK by impinging on the molecular brake and DFG latch regions, which form part of the allosteric communication network we propose in our study. However, a direct comparison of the crystal structures of inhibited and activated kinases in complex with AMP-PCP (*Mohammadi et al., 1996b*), a nonhydrolyzable ATP analog, clearly shows that ATP binding alone is ineffective in altering the hydrogen bonding network of the molecular brake or switching the phenylalanine position in the DFG latch. In other words, despite binding the ATP analog, unphosphorylated FGFRK remains in the inhibited conformation. Notably, in contrast to the fully ordered ATP analog in the crystal structure of the phosphorylated kinase-ATP complex, only the adenosine portion of the ATP analog is ordered in the crystal structure of unphosphorylated kinase (*Figure 6—figure supplement 1A*). This is due to the fact that the DFG conformation for the unphosphorylated kinase is not permissive for $Mg^{2+}$ ion coordination by the conserved aspartic acid resulting in a disordered state of the phosphate moieties within the ATP analog. It is also noteworthy that the adenosine of the bound analog exhibits a high-temperature factor further suggesting a less tightly bound conformation by the unphosphorylated kinase. Therefore, compared to pathogenic mutations that directly target the four regions of the allosteric network, ATP is a poor extrinsic effector of conformational equilibrium between the active and inhibited states.

To support these static crystallographic observations, we acquired NMR spectra aimed at probing chemical shift perturbations induced upon AMP-PCP binding in solution. Specifically, we compared the $^1H/^{13}C$ Ile/Val/Leu methyl and $^1H/^{15}N$ amide chemical shift changes upon addition of a saturating amount of AMP-PCP:$Mg^{2+}$ mix. The rationale behind comparing chemical shifts is essentially the same as that described above for probing the effect of A-loop phosphorylation. Namely, if ATP binding facilitated a conformational switch of the autoinhibited kinase to the activated state, then we would detect significant chemical shift perturbations in C-lobe residues that tether the A-loop tyrosines to the C-lobe. As expected, addition of AMP-PCP induced large chemical shift changes for residues in the proximity of the ATP binding pocket including the 'gatekeeper' residue (V564) and glycine-rich loop residues (i.e. G488, G490). By contrast, most C-lobe residues including residues V691 and L712, that tether A-loop tyrosine to the C-lobe, were completely insensitive to the presence of AMP-PCP or experienced only minor perturbations (i.e. L675) implying that nucleotide binding does not significantly impact the A-loop conformation (*Figure 6—figure supplement 1B*). As shown in *Figure 2—figure supplement 1A*, phosphorylation at Y657 induced several major chemical shift changes in the kinase C-lobe. Hence, in agreement with X-ray data, the NMR results also demonstrate that AMP-PCP (mimic of ATP) binding to an autoinhibited enzyme cannot force the A-loop to the active conformation. Therefore, compared to pathogenic mutations that directly target the four regions of the allosteric network, ATP is a relatively poor extrinsic effector of conformational switching between the active and inhibited states. Furthermore, given that ATP concentration in the cytosol is relatively constant between 1 and 10 mM, ATP is unlikely to serve as a physiological means in modulating intrinsic dynamics of the FGFRK.

## Multi-site design of allosteric network minimizes the deleterious impact of gain-of-function pathogenic mutations while creating multiple entry points for opportunistic mutations

The kinase activity data establish that none of the FGFR2K pathogenic mutants are capable of achieving the activity exerted by A-loop phosphorylation (*Figure 7*). Consistent with these activity measurements, NMR data show that the pathogenic mutations only partially mimic the active kinase conformation achieved by A-loop phosphorylation (*Chen et al., 2013*). For example, as previously observed in backbone $^1H/^{15}N$ TROSY-HSQC experiments (*Chen et al., 2013*), the $^1H/^{13}C$ HMQC

spectrum for monophosphorylated WT also showed several residues at the DFG latch with attenuated signal intensities (*Figure 11*). In contrast, the majority of pathogenic mutants showed well-resolved peaks with only a few residues having significantly diminished signal intensities. The E565A mutant showed the largest signal attenuations for backbone amide and methyl group resonances where notable reductions in peak heights were seen around the DFG motif albeit to a lesser extent than that for A-loop phosphorylated FGFR2K (*Figure 11*). This observation is consistent with the fact that the E565A mutant possesses the greatest intrinsic activity among the FGFR2K mutants studied.

The incomplete mimicry of the active state by pathogenic mutations reflects the skewed design of the multi-site allosteric network. Specifically, three out of the four sites of the allosteric network (molecular brake, DFG latch, A-loop plug) mediate strong hydrogen bonding and hydrophobic intramolecular contacts that repress the kinase from adopting an active conformation. Only the αC tether site provides van der Waals forces to encourage the active conformation. Thus, evolution of the FGFR kinase structure has favored a biased design to impose tight autoinhibition on the kinase that severely limits the ability of the enzyme to adopt the active state. Consequently, mutational derepression of only one of the three autoinhibitory sites (molecular brake, DFG latch or A-loop plug) or mutational strengthening of the αC tether alone does not completely convert the kinase to the active state, which implies that mutation in one site partially accesses the conformation of the remaining three sites. In other words, A-loop phosphorylation and the subsequent formation of an intra-loop pTyr-Arg salt bridge is the most effective way to fully couple the four sites to give rise to the largest fraction of activated kinase population. Notably, the differential abilities of pathogenic mutations to couple all four sites translate into differences in active state populations for these mutations which correlate with the gradations in gain-of-function and the associated clinical severity found in skeletal syndromes caused by the mutations (*Chen et al., 2013*).

The multi-site design of the allosteric network implies that double mutations from two sites may work in conjunction to push the kinase toward the active state. To test this hypothesis, we engineered the following double mutants that combine mutations from two different sites namely: molecular brake and A-loop (E565A/K659M), molecular brake and DFG latch (E565A/I547V and E565A/L617M), molecular brake and αC tether (E565A/D650V, and E565A/M537I), and DFG latch and αC tether (I547V/M537I and L617M/D650V) (*Table 2*). In order to assess whether strengthening the αC tether alone could overcome the repressive forces that are exerted by three autoinhibitory sites, an additional M537I/D650V double mutant was also prepared. Lastly, as a control, we constructed the N549H/E565A double mutant in which two mutations target the same allosteric site (i.e. the molecular brake). The activities of the double mutants were directly compared against those of respective single mutants as well as the unphosphorylated and phosphorylated WT kinases using a mass spectrometry-based in vitro substrate phosphorylation assay. As shown in *Figure 12A*, the activities of all double mutants targeting different allosteric regions were higher than the individual mutants. In fact, the activity of the E565A/K659M double mutant nearly reached the maximal activity achieved by A-loop phosphorylated WT FGFR2K. The D650V/M537I mutant was more active than either of the D650V and M537I single mutants as well as their added activities, but was unable to reach the activity of the A-loop phosphorylated WT kinase. By contrast, double mutants targeting the same allosteric site (N549H/E565A) or two adjoining sites (I547V/E565A) gave only a subtle increase in activity beyond those of the single mutants. This lack of synergy between N549H and E565A mutations in elevating kinase activity is expected as either mutation alone is sufficient to disrupt the hydrogen bonding network at the molecular brake. These extensive kinase assay data show that double mutations among the four sites of the allosteric network can reinforce each other in increasing the enzymatic activity. The physiological evidence for synergism between gain-of-function mutations is provided by a recent discovery of a rare case of Thanatophoric Dysplasia (TD) caused by double missense FGFR3 mutations in cis (N540K and V555M) located in the molecular brake and near the DFG latch (*Pannier et al., 2009*). While the N540K mutation accounts for 70% of Hypochondroplasia cases and results in a mild form of dwarfism, the combination of this mutation with V555M results in TD, a neonatal lethal skeletal syndrome, presumably due to greater gain-of-function of the double mutant relative to the single mutations alone.

To provide structural support for the enhanced activities of the pathogenic double mutations, we solved crystal structures for the two most active double mutants (E565A/K659M and E565A/D650V). In agreement with their elevated kinase activities, both double mutants crystallized in an activated conformation displaying an active A-loop conformation, a disengaged molecular brake, a weakened

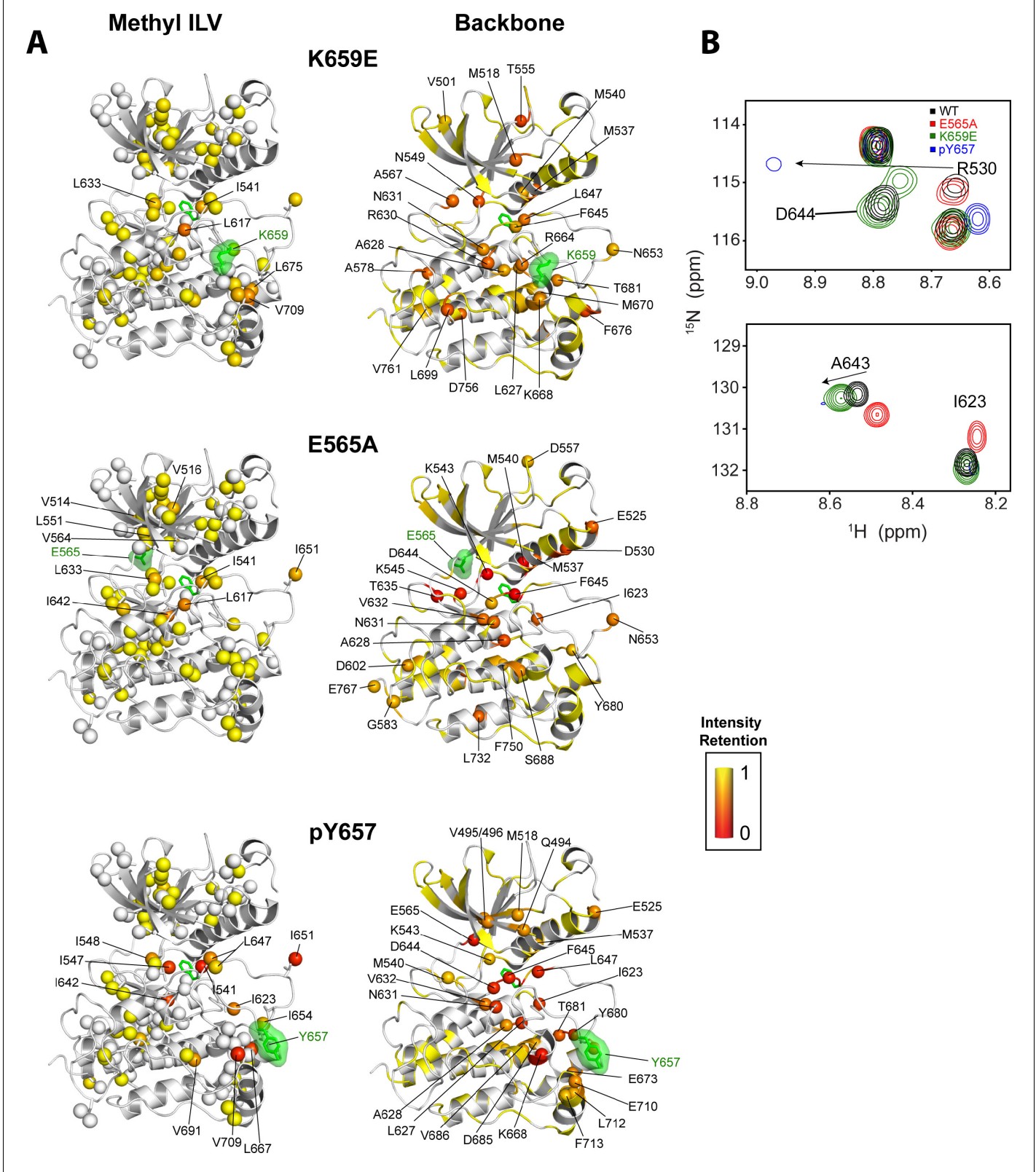

**Figure 11.** Intensity retentions of phosphorylated Y657 WT FGFR2K and pathogenic mutants K659E and E565A. (**A**) Intensity retentions from $^{1}$H/$^{13}$C HMQC and $^{1}$H/$^{15}$N TROSY-HSQC spectra for Ile, Leu, and Val methyl groups (left column) and backbone (right column) of K659E (top), E565A (middle), and phosphorylated Y657 (bottom) relative to unphosphorylated WT FGFR2K. Intensity retentions were calculated by dividing the peak intensities of

*Figure 11 continued on next page*

*Figure 11 continued*

each mutant and phosphorylated WT protein by those of the WT. The site of mutation or phosphorylation is shown as a green surface. The side chain phenylalanine (F645) in the DFG motif is also depicted with green sticks in each structure to highlight its involvement in the allosteric pathway. All marked residues on the structures corresponding to the Ile, Leu, and Val methyl group data had intensity retentions below 0.8. Similarly, all residues marked and represented as spheres for the backbone data indicate intensity retentions below 0.8. The intensity retentions are represented on a color scale (red to yellow) from 0 to 1 where 1 represents full retention and 0 represents a complete loss of signal. (**B**) Overlay of $^1$H/$^{15}$N TROSY-HSQC spectra for unphosphorylated WT, mono-phosphorylated Y657, E565A, and K659E showing the diminished or loss of peak intensities for phosphorylated Y657 and/or E565A. The top overlay highlights residue D644 within the DFG motif, which is present for WT and K659E, but missing from the phosphorylated Y657 spectrum and significantly diminished for E565A. Note that at a lower contour cutoff, D644 is observed for E565A. R630 is also shown in this same panel and indicates a large chemical shift perturbation for K659E that was previously reported for the pathogenic mutants characterized at K659 (*Chen et al., 2013*). The bottom spectral overlay in panel B shows a similar trend for residue G643. Here, signal strength for phosphorylated Y657 FGFR2K is significantly diminished, while E565A shows a chemical shift perturbation without a loss in signal intensity.

DFG latch and an engaged αC tether (*Figure 12B,C* and *Table 1*). Taken together with our extensive structural and functional results on single mutants, these data on the double mutants further support a four-site long-range allosteric network that controls FGFRK conformational dynamics and activity. Although the four-site design of the allosteric network introduces additional entry points for pathogenic mutations, we surmise it might be more advantageous over a single switch mechanism from a pathophysiological perspective. Specifically, pathogenic mutations in a one-switch mechanism involving idealized cooperativity would fully activate the kinase thus leading to more dire consequences than in the multi-site mechanism where a single mutation can only achieve partial kinase activation. Since the probability of acquired mutations simultaneously occurring at two sites in a cis fashion is extremely low, we propose that this multi-site mechanism has evolved as a defense mechanism to diminish the detrimental effects of opportunistic mutations.

## Materials and methods

### Protein expression, purification, and crystallization

The cDNA fragments encoding residues P455 to E765 of human FGFR1c (Accession code: P11362-1), P458 to E768 of human FGFR2c (Accession code: P21802-1) were amplified by PCR and subcloned into a pET bacterial expression vector with an NH$_2$-terminal 6XHis-tag to aid in protein purification. All the mutations were introduced using the QuikChange site-directed mutagenesis kit (Stratagene). The bacterial strain BL21(DE3) cells were transformed with the expression constructs, and kinase expression was induced with 1 mM isopropyl-L-thio-B-D-galactopyranoside overnight at 16 to25°C depending on the construct. The cells were lysed, and the soluble kinase proteins were purified according to the published protocol (*Chen et al., 2007*). Traces of phosphorylation on WT and mutant kinases were removed by treating the proteins with FastAP Thermosensitive Alkaline Phosphatase (Thermo Scientific) and phosphorylation-free samples were repurified by anion exchange chromatography (Mono Q, GE Healthcare Life Sciences). The substrate peptide corresponds to the C-terminal tail of FGFR2 and contains five authentic tyrosine phosphorylation sites (Y769, Y779, Y783, Y805, and Y812). Isotopically labeled WT and mutant kinases were expressed and purified similar to their unlabeled counterparts.

### Protein crystallization

The purified FGFR2 kinase mutants were concentrated to about 10–100 mg/ml using Amicon Ultra-4 10K Centrifugal Filters (Millipore). Prior to crystallization, the pathogenic kinases were mixed with ATP-analogue (AMP-PCP) and MgCl$_2$ at a molar ratio of 1:3:15. Initial crystals of pathogenic kinases were grown by hanging drop vapor diffusion at 20°C using crystallization buffer composed of 25 mM HEPES pH7.5, 15%–25% w/v PEG 4000, 0.2–0.3 M Ammonium Sulfate and were further optimized by appropriate additives.

### X-ray data collection and structure determination

Diffraction data were collected on single cryo-cooled crystals at beamlines X-4A and X-4C at the National Synchrotron Light Source, Brookhaven National Laboratory. Crystals were stabilized in

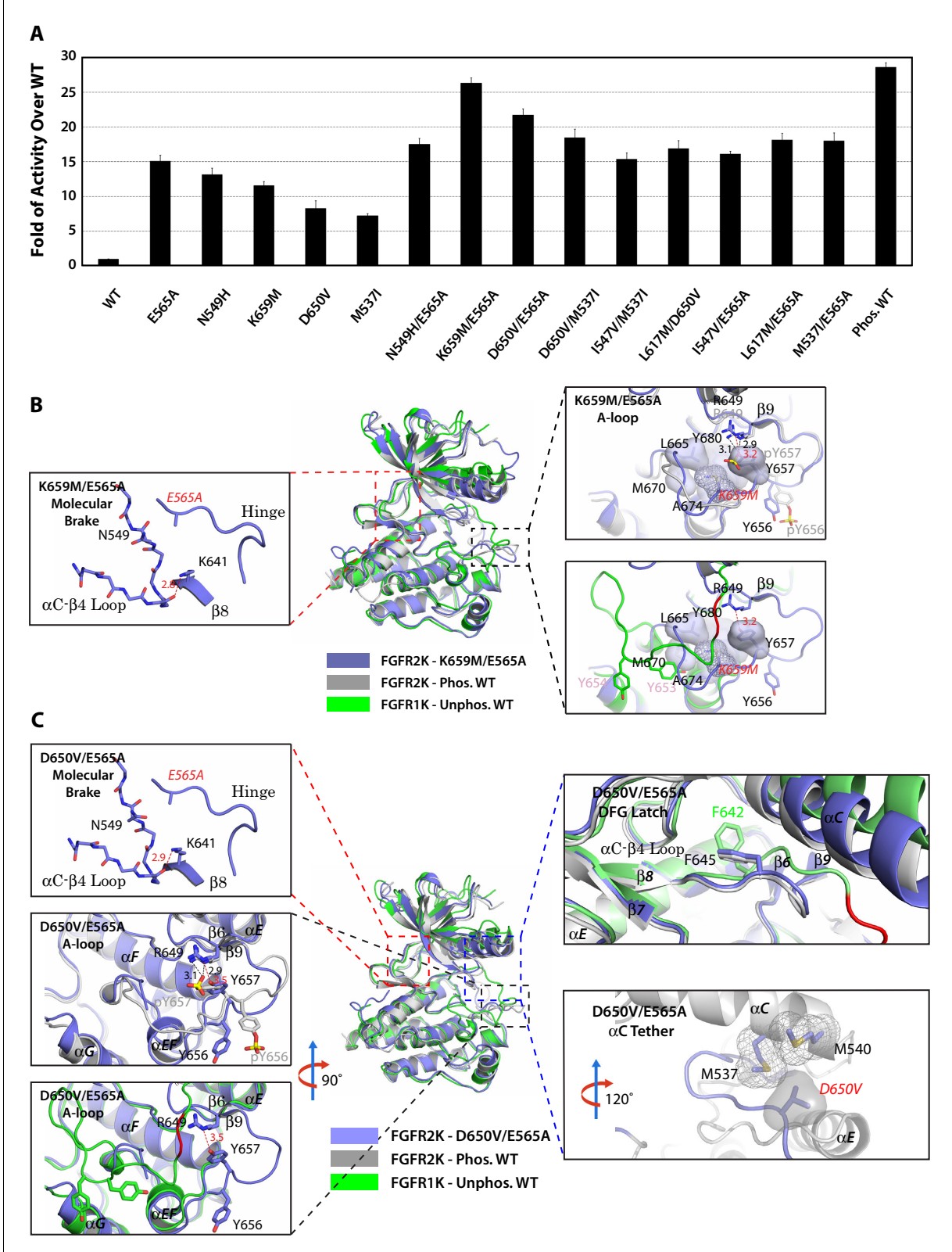

**Figure 12.** Combined mutations from two sites of the allosteric network further stabilize the active state conformation of FGFR2K. (**A**) Comparison of substrate phosphorylation activities of nine double mutants that carry combined mutations from the same site (N549H/E565A) or two different sites. Activities were measured 30 s post reaction and are expressed as the fold of activity over the unphosphorylated WT FGFR2K. Error bars represent mean ± SD. (**B**) Superimposition of the crystal structures of the K659M/E565A double mutant (in blue), unphosphorylated autoinhibited FGFR1K (PDB ID:

*Figure 12 continued on next page*

*Figure 12 continued*

1FGK [*Mohammadi et al., 1996*], in green), and the A-loop tyrosine phosphorylated WT FGFR2K (PDB ID: 2PVF [*Chen et al., 2007*], in grey). The right inset shows a zoomed-in view of the molecular brake region of the K659M/E565A double mutant. Note that reminiscent of the A-loop phosphorylated activated FGFR2K, the molecular brake is completely disengaged in the unphosphorylated K659M/E565A double mutant. The upper inset to the right of the panel shows a zoomed-in view of the A-loop conformation of the K659M/E565A double mutant overlaid onto that of A-loop phosphorylated FGFR2K. Lower inset to the right of this panel shows a zoomed-in view of the A-loop conformation of K659M/E565A double mutant superimposed onto that of unphosphorylated autoinhibited WT FGFR1K. In the lower inset, the section of A-loop in the unphosphorylated FGFR1K that is destined to become *β*9 strand upon A-loop phosphorylation is highlighted in red. Note that the A-loop in the K659M/E565A double mutant is in an activated conformation. The hydrophobic interactions that the K659M mutation introduces to support the active conformation of the A-loop are represented by semitransparent surfaces, and the hydrogen bond between Y657 and R649 is shown as a red dashed line with the distance given in Å. (C) Comparison of the crystal structures of the D650V/E565A double mutant (in blue), unphosphorylated autoinhibited FGFR1K (PDB ID: 1FGK [*Mohammadi et al., 1996*], in green), and the A-loop tyrosine phosphorylated WT FGFR2K (PDB ID: 2PVF [*Chen et al., 2007*], in grey). The top left inset shows a close-up view of the molecular brake region of the D650V/E565A double mutant. Note that reminiscent of the A-loop phosphorylated activated FGFR2K, the molecular brake is disengaged in the unphosphorylated double mutant. The middle inset on the left shows the superimposition of the A-loop region for unphosphorylated D650V/E565A and A-loop tyrosine phosphorylated WT FGFR2K. The bottom inset to the left shows the superimposition of the A-loop region for unphosphorylated D650V/E565A and unphosphorylated inhibited WT FGFR1K. Note that the A-loop in the D650V/E565A double mutant is primarily in an activated conformation. The upper inset to the right shows a close-up view of the DFG latch region showing that the DFG phenylalanine in the D650V/E565A double mutant adopts the same rotamer position as the corresponding phenylalanine in the A-loop tyrosine phosphorylated WT FGFR2K. The lower inset to the right shows a close-up view of the αC tether where hydrophobic contacts are made between V650 and the two methionines in the αC helix. The hydrophobic interactions are represented by mesh and semitransparent solid surface, and the hydrogen bond between Y657 and R649 is shown as a black dashed line with the distance given in Å. In panels **B** and **C**, the labels for the mutant kinases are in red. Only side chains of relevant residues are rendered in sticks. Atom colorings are as in *Figure 1*.

mother liquor by stepwise increasing glycerol concentration to 20%, and then flash-frozen in liquid nitrogen. All diffraction data were processed using *HKL2000 Suite* (*Otwinowski and Minor, 1997*). Molecular replacement solutions were obtained with *AmoRe* (*Navaza, 1994*) using the FGFR2 kinase structure (PDB ID: 2PVY [*Chen et al., 2007*]) as the search model. Model building was carried out using *Coot* (*Emsley et al., 2010*) (RRID:SCR_014222) and iterative refinements were completed using *PHENIX* (*Adams et al., 2010*) (RRID:SCR_014224). Atomic superimpositions were made using *lsqkab* (*Kabsch, 1976*) in the *CCP4 Suite* (*Collaborative Computational Project, Number 4, 1994*) (RRID:SCR_007255) and structural representations were prepared using *PyMol* (*DeLano, 2002*) (RRID:SCR_000305).

## Kinase substrate phosphorylation assay

Unphosphorylated WT and mutant FGFR2 kinases as well as phosphorylated WT FGFR2 kinase were mixed with kinase reaction buffer containing ATP, $MgCl_2$ and the substrate peptide to the final concentrations of 13.5 µM (kinase), 262 µM (substrate), 10 mM (ATP), and 20 mM ($MgCl_2$). The reactions were quenched at different time points by adding EDTA to the reaction mix at the final concentration of 33 mM. The progress of the substrate phosphorylation was monitored by native-PAGE, and the phosphate incorporation into the substrate peptide was quantified by time-resolved MALDI-TOF mass spectrometry by comparing signals from phosphorylated and the cognate non-phosphorylated peptides as previously published (*Chen et al., 2013*).

## NMR spectroscopy

All NMR experiments were carried out at a temperature of 25°C using the FGFR2K isoform at [1]H frequencies of 600 MHz (Bruker *AVANCE III* spectrometer equipped with Z-axis gradient TCI Cryo-Probe) or 800 MHz (Bruker *AVANCE III HD* spectrometer equipped with Z-axis gradient TCI CryoProbe). Single-site mutants were prepared at 300 µM in a fully unphosphorylated form by treating with alkaline phosphatase similar to those samples used for crystallization. The NMR buffer for all samples was 25 mM HEPES and 150 mM NaCl at a pH of 7.5. The assignments for the single site mutants were transferred from that determined previously for unphosphorylated wild-type FGFR2K (*Chen et al., 2013*). Combined chemical shift perturbations obtained from [1]H/[15]N heteronuclear correlation experiments were calculated from the equation below:

$$\Delta\delta = \sqrt{(0.154\Delta\delta_N)^2 + \Delta\delta_H^2} \tag{1}$$

The assignment of Ile $C_{\delta 1}$, Val $C_{\gamma 1/2}$, and Leu $C_{\delta 1/2}$ methyl sites was accomplished by using 3D experiments that correlated the methyl $^1H/^{13}C$ shift with the CA, CB, and CO chemical shifts in two separate 3D datasets (*Tugarinov and Kay, 2003*). These chemical shifts were matched with those previously assigned using TROSY backbone triple resonance experiments on wild-type FGFR2K (*Chen et al., 2013*). Using this approach, we were able to unambiguously assign >90% of Ile, Val, and Leu residues. Combined chemical shift perturbations determined from $^1H/^{13}C$ heteronuclear experiments were calculated from the following equation:

$$\Delta\delta = \sqrt{(0.25\Delta\delta_C)^2 + \Delta\delta_H^2} \qquad (2)$$

Generation of a phosphorylated tyrosine at position Y657 was achieved by mutating all other phosphorylatable tyrosine residues in unphosphorylated FGFR2K (Y466F, Y586L, Y588L, Y656F). Next the protein was concentrated to 800 µM and ATP and $MgCl_2$ were added to a final concentration of 20 mM and 30 mM, respectively. After 10 min at room temperature, the reaction was injected onto a size exclusion column to remove ATP and $MgCl_2$, and then subsequently injected onto an anion exchange column to remove traces of unphosphorylated kinase.

PRE experiments were carried out on a selectively labeled $^{15}N$ tyrosine I707C mutant of FGFR2K conjugated with an MTSL spin label. MTSL labeling was carried out as previously described (*Huang et al., 2016*). A $^1H/^{15}N$ HSQC spectrum was acquired for both the oxidized and reduced forms of the MTSL spin label. The oxidized spin label was reduced using 10 molar equivalents of ascorbic acid. The intensity retention was calculated by dividing the peak heights of the oxidized sample by those of the reduced sample, and distances were calculated as previously described using a correlation time of 22.2 ns (*Battiste and Wagner, 2000*).

CPMG relaxation dispersion experiments were carried out for unphosphorylated FGFR2K and the K659E single-site mutant. $^{13}C$ Ile, Leu, and Val multiple quantum (*Korzhnev et al., 2004b*) and $^{15}N$ backbone TROSY single quantum (*Loria et al., 1999*) dispersion curves were collected for wild-type FGFR2K at 600 MHz with the following $\nu_{CPMG}$ frequencies in duplicate: 50, 100, 150, 200, 250, 400, 600, 800, and 1000 Hz. Similarly K659E curves were acquired at 600 MHz and 800 MHz for methyl groups and 600 MHz for backbone $^{15}N$ using the same range of CPMG frequencies as for WT FGFR2K. A constant time period of 40 msec was used for all CPMG experiments (*Mulder et al., 2001*). Effective $R_2$ values ($R_{2eff}$) were calculated using *Equation 3*:

$$R_{2eff} = -ln\left(\frac{I_0}{I_{\nu,CPMG}}\right)\frac{1}{T} \qquad (3)$$

where T is the value of the constant time delay, $I_{\nu,CPMG}$ is the intensity of the peak corresponding to the CPMG frequency, and $I_0$ is the intensity of the reference spectrum with no constant time delay. The $\Delta R_{2,eff}$ value is defined to be the difference between the 50 Hz and 1000 Hz points and is an indicator of the amount of dispersion in the CPMG experiment. In order to derive the exchange rate ($k_{ex}$) for the conformational change in K659E, the methyl CPMG dispersion curves were fit with the fast-exchange Luz-Meiboom equation given below assuming no contribution from $^1H$ chemical shift dispersion ($\Delta\omega_{1H} = 0$) (*Luz and Meiboom, 1963*):

$$R_{2eff} = R_2 + \frac{p_A p_B \Delta\omega^2}{k_{ex}}\left(1 - \frac{4\nu_{CPMG}}{k_{ex}}tanh\left(\frac{k_{ex}}{4\nu_{CPMG}}\right)\right) \qquad (4)$$

In total, six residues showed significant dispersions at both 600 MHz and 800 MHz spectrometers and were fit in a global fashion to yield a single $k_{ex}$ value (I548, I623, L627, L647, I707, V709). Individual fits to $k_{ex}$ were also carried out using *equation 3* and were found to give values ranging between 1400 and 2500 s$^{-1}$. The similarity of the individual $k_{ex}$ fits was used as justification for the global fits. Note that the error bar of the globally fit value reflects the standard deviation of the individual fits to $k_{ex}$.

AMP-PCP binding experiments were carried out by adding 25 mM AMP-PCP and 10 mM $MgCl_2$ to unphosphorylated FGFR2K of $^{13}C$ ILV methyl and $^{15}N$ amide labeled FGFR2K (300 µM). $^1H/^{15}N$ TROSY-HSQC and $^1H/^{13}C$ HSQC experiments were acquired in both the presence and absence of AMP-PCP/$Mg^{2+}$. All NMR data were processed with *NMRPipe* (*Delaglio et al., 1995*) and analyzed using *Sparky* (*Goddard and Kneller, 1997*) (RRID:SCR_014228).

## CHESCA analysis

Complete linkage CHESCA was carried out as described by Melacini and co-workers (*Selvaratnam et al., 2011*; *Boulton et al., 2014*). The chemical shifts from a series of mutations at K659 (T, N, Q, M, E) and wild-type phosphorylated and unphosphorylated FGFR2K were used as input. Residues with chemical shift differences between phosphorylated and unphosphorylated FGFR2K less than 10 Hz in $^1$H and 5 Hz in $^{15}$N were excluded from the analysis. The combined chemical shift (CCS) was calculated by summing the $^{15}$N and $^1$H chemical shifts with a 0.2 scaling factor for $^{15}$N (CCS = $0.2\delta_N + \delta_H$). The clustering was carried out with the software package Cluster 3.0 (http://bonsai.hgc.jp/~mdehoon/software/cluster) (*de Hoon et al., 2004*). Dendrograms were prepared using Java TreeView (http://jtreeview.sourceforge.net) (*Saldanha, 2004*). Residue clusters were selected that had correlation coefficients $|r_{ij}| > 0.97$. In order to determine whether these residues belonged to the same functional network, a chemical shift sub-matrix was constructed only of residues within the identified clusters. The resulting state-based dendrograms are shown in *Figure 9—figure supplement 2* and reveal that the three displayed clusters in *Figure 9D* belong to the same functional network based on kinase activity assays carried out previously (*Chen et al., 2013*).

## Acknowledgements

This work was supported by the NIDCR grant DE13686 (to MM), NINDS grant P30 NS050276 (to TAN), and NIAID grant R01AI108889 (to NJT). Beamlines X-4A and X-4C at the National Synchrotron Light Source, Brookhaven National Laboratory (RRID:SCR_011123), a DOE facility, are supported by New York Structural Biology Consortium. The NMR data collected at NYU were supported by an NIH S10 grant (OD016343) while those NMR data acquired at the New York Structural Biology Center were supported by grants from NYSTAR and NIH (S10OD016432). The authors thank Prof. Arthur Palmer and Dr. Ying Li for sharing the backbone $^{15}$N TROSY-CPMG relaxation dispersion pulse sequence code for Bruker spectrometers.

## Additional information

### Funding

| Funder | Grant reference number | Author |
|---|---|---|
| National Institute of Dental and Craniofacial Research | DE13686 | Moosa Mohammadi |
| National Institute of Neurological Disorders and Stroke | P30 NS050276 | Thomas A Neubert |
| National Institute of Allergy and Infectious Diseases | R01AI108889 | Nathaniel J Traaseth |

The funders had no role in study design, data collection and interpretation, or the decision to submit the work for publication.

### Author contributions

HC, Conceptualization, Data curation, Software, Formal analysis, Validation, Investigation, Visualization, Methodology, Writing—original draft, Project administration, Writing—review and editing; WMM, Data curation, Formal analysis, Validation, Investigation, Visualization, Writing—original draft, Writing—review and editing; M-KC, ZH, JD, SPB, Data curation, Formal analysis; WG, SB, Resources, Data curation; TAN, Supervision, Funding acquisition, Investigation, Writing—review and editing; NJT, MM, Conceptualization, Resources, Data curation, Software, Formal analysis, Supervision, Funding acquisition, Validation, Investigation, Visualization, Methodology, Writing—original draft, Project administration, Writing—review and editing

### Author ORCIDs

Moosa Mohammadi, http://orcid.org/0000-0003-2434-9437

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
