## [Decision Letter]

Thank you for submitting your article "Elucidation of a Four-Switch Allosteric Network in FGF Receptor Tyrosine Kinases" for consideration by *eLife*. Your article has been favorably evaluated by Tony Hunter (Senior Editor) and three reviewers, one of whom is a member of our Board of Reviewing Editors. The reviewers have opted to remain anonymous.

The reviewers have discussed the reviews with one another and the Reviewing Editor has drafted this decision to help you prepare a revised submission.

Summary:

This work builds upon the authors' previous structural analyses of the FGFR kinase domain and identifies a set of intra-protein interactions important in determining whether a FGFR kinase domain adopts the active or the inactive conformation, Based on extensive crystallographic and biochemical data, as well as technically challenging NMR experiments, it shows that the thermodynamic components underlying FGFR regulation are embedded in the kinase domain in a distributed fashion. Of particular interest is the finding that in the active conformation many regions of a FGFR kinase, including the DFG motif, is more flexible than in the inactive conformation.

Essential revisions:

1) While this paper is primarily concerned with autoinhibition *FGFR2* kinase, a definitive crystal structure of the auto-inhibited kinae remains unavailable. The only one discussed in the manuscript (PDB: 2PSQ) was thought to be complicated by crystal lattice packing by authors. Although the functional data and NMR data implies that FGFR2K, like FGFR1K, is autoinhibited in the unphosphorylated state, no direct evidence indicates that, as this study assumes, the autoinhibited structure of FGFR2K is necessarily highly similar to the other FGFR kinases. This is a potentially important weakness of this study, especially because allosteric communication can be sensitive to subtle structural differences and vary significantly among homologue proteins. In the revision the authors need to address this issue with new data or with additional discussions.

2) Presumably in addition to changes in residue packing in key regions, ATP binding, which occurs between the molecule brake and the DFG motif, may also affect the balance between the active and the inactive conformations of FGFR2K. This should be addressed in the revision.

3) Figure 7 shows that different members in the FGFR family respond differently to mutations. This raises doubts as to whether the findings with FGFR2K will be generally applicable across all members of this family protein.

4) The reviewers take issue with the choice of the key terms the manuscript uses. Specifically, the reviewers question the usage of the terms "allosteric networks" and "switches" in describing the findings. By way of analogy, one can assume two proteins bind through an extensive interface and a mutation at the interface that abolishes the binding. This would disrupt the interface interactions, including those involving residues distal to the mutation site, but it would be inappropriate to describe the binding interface as an allosteric network and the mutated residues as switches, despite the distal affects. The reviewers think, for similar reasons, the two terms may not be appropriate to describe this work, which identifies key structural determinants involved in FGFR autoinhibition and activation. It appears that the mutational data (including those that have pathogenic effects and those that are rationally introduced by authors) can also be explained just by the structure features captured in crystal structures, without invoking the concept of a "four switch allosteric network". DFG, α-C and activation loop are already well known important elements for kinase activation. The reviewers suggest that the authors explain why the usage of these terms are necessary or use different terms.

---

## [Author Response]

[…]

*Essential revisions:*

*1) While this paper is primarily concerned with autoinhibition FGFR2 kinase, a definitive crystal structure of the auto-inhibited kinae remains unavailable. The only one discussed in the manuscript (PDB: 2PSQ) was thought to be complicated by crystal lattice packing by authors. Although the functional data and NMR data implies that FGFR2K, like FGFR1K, is autoinhibited in the unphosphorylated state, no direct evidence indicates that, as this study assumes, the autoinhibited structure of FGFR2K is necessarily highly similar to the other FGFR kinases. This is a potentially important weakness of this study, especially because allosteric communication can be sensitive to subtle structural differences and vary significantly among homologue proteins. In the revision the authors need to address this issue with new data or with additional discussions.*

We agree with the reviewers that ideally we should solve a crystal structure of unphosphorylated FGFR2K in different space groups to eliminate the bias introduced by the lattice contacts on the A-loop conformation in *2PSQ*. However, despite numerous attempts, we failed to crystallize the unphosphorylated FGFR2K in a different space group. We would like to mention that other groups have faced the same problem^1^. Evidently, strong lattice contact forces prevail and conceal a truly autoinhibited conformation of the A-loop in FGFR2K. To overcome this problem, we decided to carry out additional NMR experiments to assess the conformation of the A-loop of FGFR2K in solution (free of lattice contacts). As discussed in the manuscript (Results and Discussion), the conformations of A-loop in both the crystal structures of unphosphorylated FGFR1K (3KY2) and FGFR4K (4QQT) are essentially identical and both show the A-loop tyrosines to pack tightly against the C-lobe. In stark contrast, in the FGFR2K structure (2PSQ) the N-terminal half of A-loop adopts an active-like conformation while the C-terminal half of A-loop adopts an alternative conformation (neither inhibited nor active). Importantly, in 2PSQ, the A-loop tyrosine residues face away from the C-lobe and are completely solvent exposed. Based on these crystallographic data, we devised two NMR experiments to obtain proximity information between the A-loop tyrosines and the C-lobe in the unphosphorylated FGFR2K in solution that would help resolve ambiguity regarding the A-loop conformation:

A) In the first approach, we analyzed the chemical shift differences between the unphosphorylated and monophosphorylated (pY657) activated FGFR2K by plotting them onto the crystal structures of unphosphorylated FGFR1K (3KY2) and FGFR2K (2PSQ). We reasoned that if the A-loop tyrosines in the unphosphorylated FGFR2K were interacting with the C-lobe as observed in the unphosphorylated FGFR1K and FGFR4K structures, then we would detect chemical shift differences upon A-loop phosphorylation for the A-loop tyrosines and for the C-lobe residues that pack against the A-loop tyrosines. As shown in Figure 2—figure supplement 1, we plotted the chemical shift perturbations onto FGFR1K and FGFR2K structures and found substantial perturbations that clustered in a region directly surrounding the location of the A-loop tyrosines in the FGFR1K structure (i.e., catalytic loop, αEF, αF, αG helices). These NMR results cannot be reconciled with the observed conformation of the A-loop in the FGFR2K structure (2PSQ) as the A-loop tyrosines in this structure do not interact with these perturbed residues in the C-lobe. This NMR analysis supports the notion that the unphosphorylated FGFR2K structure (2PSQ) represents a partially activated conformation due to crystal packing.

B) In the second approach, we carried out new NMR experiments utilizing the paramagnetic relaxation enhancement (PRE) technique to directly measure distances between A-loop tyrosine residues and the kinase C-lobe. PRE effects are strong up to a distance of ~20 Å but diminish at distances greater than ~25 Å. PRE experiments require a free cysteine for covalently linking a paramagnetic spin label onto the protein. To this end we selected I707 of FGFR2K for introducing a free cysteine since this isoleucine is positioned ideally to dissect the A-loop conformational difference between unphosphorylated FGFR2K (2PSQ) and FGFR1K/FGFR4K (3KY2/4QQT) structures (Figure 2—figure supplement 1). In the unphosphorylated FGFR2K structure (2PSQ), the Cγ of I707 and the nitrogen atom of Y656 are ~27 Å apart, whereas the corresponding distance between V704 in Y653 in FGFR1K (3KY2) is much shorter (~15 Å). To carry out these experiments with good confidence and resolution, the I707C FGFR2K mutant was selectively labeled with ^15^N tyrosine prior to covalently attaching an MTSL label. ^1^H/^15^N HSQC experiments on the mutant sample in the absence and presence of reducing agent showed notable reductions in peak intensities for the two A-loop tyrosines in the oxidized samples that are consistent with a distance less than 20 Å (Figure 2—figure supplement 1). We estimated the distance from the intensity retention using the approach of Battiste and Wagner and calculated a value of ~19 Å. While this is slightly longer than the distance measured in the FGFR1K crystal structure (3KY2), it is within the acceptable range of experimental uncertainty stemming from the packing of MTSL against the protein (i.e., possible side-chain rotamers). Hence, as with the chemical shift data, our PRE results also imply that the A-loop conformation in the unphosphorylated FGFR2K structure (2PSQ) is not a genuinely inhibited conformation, and that in solution the A-loop of FGFR2K likely adopts a similar conformation as that displayed by the A-loops of unphosphorylated FGFR1K (3KY2) and FGFR4K (4QQT) in the crystal.

*2) Presumably in addition to changes in residue packing in key regions, ATP binding, which occurs between the molecule brake and the DFG motif, may also affect the balance between the active and the inactive conformations of FGFR2K. This should be addressed in the revision.*

As noted by the reviewer, binding of ATP into the cleft between the N- and C-lobes of the kinase involves residues from the molecular brake and the conserved DFG motif. Notably the amino group of the ATP adenine ring makes a direct hydrogen bond with the backbone carbonyl oxygen of an FGFR-invariant glutamic acid whose side chain is a central mediator of the hydrogen bonding network at the molecular brake. Moreover, the conserved aspartic acid from the tyrosine kinase-invariant DFG motif participates in ATP binding by coordinating a Mg^2+^ ion in concert with the α and β phosphate groups of ATP. Based on these structural observations, we concur with the reviewer that ATP binding may potentially promote the active state conformation of FGFR kinase by impinging on the molecular brake and DFG latch regions which form part of the allosteric communication network we propose in our study. However, a direct comparison of the crystal structures of inhibited and activated kinases in complex with AMP-PCP, a nonhydrolyzable ATP analog, clearly shows that ATP binding alone is ineffective in altering the hydrogen bonding network of the molecular brake or switching the phenylalanine position in the DFG latch^[43]^. In other words, despite binding to the ATP analog, the unphosphorylated FGFR kinase remains in the inhibited conformation. Notably, in contrast to the fully ordered ATP analog in the crystal structure of the phosphorylated kinase-ATP complex, only the adenosine portion of the ATP analog is ordered in the crystal structure of unphosphorylated kinase (Figure 6—figure supplement 1). This is due to the fact that the DFG conformation for the unphosphorylated kinase is not permissive for Mg^2+^ ion coordination by the conserved aspartic acid resulting in a disordered state of the phosphate moieties within the ATP analog. It is also noteworthy that the adenosine ring of the bound analog exhibits a high temperature factor further suggesting a less tightly bound conformation by the unphosphorylated kinase.

To support these static crystallographic observations, we acquired new NMR spectra aimed at probing chemical shift perturbations induced upon AMP-PCP binding in solution. Specifically, we compared the ^1^H/^13^C Ile/Val/Leu methyl and ^1^H/^15^N amide chemical shift changes upon addition of a saturating amount of AMP-PCP:Mg^2+^ mix. The rationale behind comparing chemical shifts is essentially the same as that described above for probing the effect of A-loop phosphorylation. Namely, if ATP binding facilitated a conformational switch of the autoinhibited kinase to the activated state, then we would detect significant chemical shift perturbations in C-lobe residues that tether the A-loop tyrosines to the C-lobe. As expected, addition of AMP-PCP induced large chemical shift changes for residues in the proximity of the ATP binding pocket including the “gatekeeper” residue (V564) and glycine-rich loop residues (i.e., G488, G490). By contrast, most C-lobe residues including residues V691 and L712, that tether A-loop tyrosine to the C-lobe, were completely insensitive to the presence of AMP-PCP or experienced only minor perturbations (i.e., L675) implying that nucleotide binding does not significantly impact the A-loop conformation (Figure 6—figure supplement 1). As shown in Figure 2—figure supplement 1, phosphorylation at Y657 induced several major chemical shift changes in the kinase C-lobe. Hence, in agreement with X-ray data, the NMR results also demonstrate that AMP-PCP (mimic of ATP) binding to an autoinhibited enzyme cannot force the A-loop to the active conformation. Therefore, compared to pathogenic mutations that directly target the four regions of the allosteric network, ATP is a relatively poor extrinsic effector of conformational switching between the active and inhibited states. Furthermore, given that ATP concentration in the cytosol is relatively constant between 1-10 mM, ATP is unlikely to serve as a physiological means in modulating intrinsic dynamics of the FGFR kinase.

*3) Figure 7 shows that different members in the FGFR family respond differently to mutations. This raises doubts as to whether the findings with FGFR2K will be generally applicable across all members of this family protein.*

We are a bit puzzled by this comment because all the data in Figure 7 including those in the inset pertain to the FGFR2K isoform only. Specifically, the data in the main part of 7B show the activities of point mutations in FGFR2K that strengthen the αC tether and lead to gain-of-function while the data in the inset of Figure 7 show the activities of the FGFR2K mutations that weaken the αC tether and lead to loss-of-function. These functional data are congruent with our hypothesis that residues with smaller side chains and hence weaker van der Waals packing potential (e.g., D650G, M537A, M540A) lead to poorer αC tether formation, while larger residue substitutions such as D650L and M537I are better adept at forming the αC tether and stabilizing the activated form of FGFR2K.

So perhaps the reviewers were referring to Figure 7. The data in Figure 7 contain activities of FGFR1K (in the inset) and FGFR2K (in the main part) isoforms. The activity data for I544V and L614V of FGFR1K shown in the inset of Figure 7 should be directly compared to those for I547V and L617V of FGFR2K in the main part of the panel. Consistent with the conservation of the allosteric network among FGFR isoforms, similar mutations in FGFR1K and FGFR2K both lead to kinase activation by influencing the DFG latch. While there is a two-fold difference in activity between L617V of FGFR2K and L614V of FGFR1K, we emphasize that the only appropriate conclusion to be drawn is that both mutations are activating. Analyzing quantitative differences among isoforms would have to factor in additional sequence differences that might favor one amino acid substitution over the other. This is a challenging endeavor which we plan to address in future studies focused on delineating subtle differences in the composition of the allosteric network that give rise to subtle differential activities of the four FGFR isoforms. Instead we only comment on the activating nature of the mutations, which is consistent with the shared DFG latch hypothesis among different FGFRK isoforms. To avoid any potential confusion to the readers, in the revised version of the manuscript we have stated explicitly that the inset data in Figure 7 correspond to loss-of-function FGFR2K mutations and those inset data in Figure 7 correspond to FGFR1K mutations.

*4) The reviewers take issue with the choice of the key terms the manuscript uses. Specifically, the reviewers question the usage of the terms "allosteric networks" and "switches" in describing the findings. By way of analogy, one can assume two proteins bind through an extensive interface and a mutation at the interface that abolishes the binding. This would disrupt the interface interactions, including those involving residues distal to the mutation site, but it would be inappropriate to describe the binding interface as an allosteric network and the mutated residues as switches, despite the distal affects. The reviewers think, for similar reasons, the two terms may not be appropriate to describe this work, which identifies key structural determinants involved in FGFR autoinhibition and activation. It appears that the mutational data (including those that have pathogenic effects and those that are rationally introduced by authors) can also be explained just by the structure features captured in crystal structures, without invoking the concept of a "four switch allosteric network". DFG, α-C and activation loop are already well known important elements for kinase activation. The reviewers suggest that the authors explain why the usage of these terms are necessary or use different terms.*

The term allostery is commonly used to describe intramolecular long-range structural or dynamic changes between remote sites in proteins that control functional conformational equilibria in protein. This usage of the allostery term has been consistently employed over the last decade (see references: Motlagh et al. 2014, The ensemble nature of allostery, Nature 508, 331-339; Hilser 2010, An ensemble view of allostery, Science 327, 653-654; Kar et al. 2010, Allostery and population shift in drug discovery, Curr Opin Pharmacol 10, 715-722; Tsai and Nussinov 2014, A unified view of “how allostery works”, PLoS Comput Biol 10, e1003394) and was stated concisely in a recent Commentary published in *PNAS* by Melacini et al. (2016, *PNAS* 113, 9407-9):

“One of the broadest definitions of allostery is in terms of long-range couplings between remote sites within a molecular system.”

In this manuscript, we used a combination of structural (X-ray and NMR) and functional methods to characterize a set of naturally selected pathogenic gain-of-function mutations allowing us to identify four dynamically coupled regions in FGFR kinase that form a long-range allosteric conduit. These four regions are involved in regulating the kinase conformational exchange between activated and inhibited states. We describe NMR data showing that structural perturbations induced by a pathogenic gain-of-function mutation in one region propagate to the remaining sites to affect the composition/packing and ultimately alter the ratio of inactive:active kinase equilibrium. Hence, we believe that usage of the term “allosteric network” is justified and consistent with published literature.

With regard to the reviewers’ analogy with interfacial interactions between two proteins, we agree that the entire surface does not automatically constitute an allosteric network. However, if a mutation within one position on the surface caused an intramolecular conformational change at a distant site on the surface in the same protein, this may constitute an interconnected allosteric network provided that the conformational change at the distant site further reduced the binding affinity in addition to the effect of the mutation. We think this refined analogy is a more apt comparison with what we have observed in FGFR kinase. Indeed, we measured the effects of distant structural perturbations within the monomeric enzyme using NMR and showed using kinase assays that double mutations gave an additive functional effect, which is consistent with long-range communication (i.e., an allosteric network). With regard to the other term, “switch”, we agree with the reviewers that this terminology may be unconventional and we have replaced it with “site” in the revised manuscript.

As to the DFG-latch and αC tether terminology, we are not using these terms to refer to the canonical DFG motif and αC helix which are well known elements in kinase activation as the reviewer mentioned. Rather, these terms are used to describe two previously unrecognized clusters of interactive residues that cooperate to modulate kinase dynamics. The DFG latch refers to a network of hydrophobic contacts centered around the phenylalanine of the DFG motif that discourages the rotation/movement of the αC helix towards the C-lobe of the enzyme. Conversely, the αC tether refers to a set of hydrophobic/van der Waals contacts between hydrophobic residues on the αC helix and the A-loop that encourages rotation/movement of the αC helix towards the C-lobe. Neither of these two clusters were known in the field, which we discovered by studying the effects of naturally occurring gain-of-function mutations in FGFR kinases. We have now revised the text to better highlight our discovery of these two new allosteric sites.

References:

1) Eathiraj, S. et al. A novel mode of protein kinase inhibition exploiting hydrophobic motifs of autoinhibited kinases: discovery of ATP-independent inhibitors of fibroblast growth factor receptor. J Biol Chem 286, 20677-87 (2011).

2) Mohammadi, M., Schlessinger, J. & Hubbard, S.R. Structure of the FGF receptor tyrosine kinase domain reveals a novel autoinhibitory mechanism. Cell 86, 577-87 (1996).